# Autophagy adaptors mediate Parkin-dependent mitophagy by forming sheet-like liquid condensates

Zi Yang[1,6], Saori R Yoshii[1], Yuji Sakai [1,2,3,7], Jing Zhang[1], Haruka Chino[1,8], Roland L Knorr[1,4,9] & Noboru Mizushima [1,5]✉

## Abstract

During PINK1- and Parkin-mediated mitophagy, autophagy adaptors are recruited to damaged mitochondria to promote their selective degradation. Autophagy adaptors such as optineurin (OPTN) and NDP52 facilitate mitophagy by recruiting the autophagy-initiation machinery, and assisting engulfment of damaged mitochondria through binding to ubiquitinated mitochondrial proteins and autophagosomal ATG8 family proteins. Here, we demonstrate that OPTN and NDP52 form sheet-like phase-separated condensates with liquid-like properties on the surface of ubiquitinated mitochondria. The dynamic and liquid-like nature of OPTN condensates is important for mitophagy activity, because reducing the fluidity of OPTN-ubiquitin condensates suppresses the recruitment of ATG9 vesicles and impairs mitophagy. Based on these results, we propose a dynamic liquid-like, rather than a stoichiometric, model of autophagy adaptors to explain the interactions between autophagic membranes (i.e., ATG9 vesicles and isolation membranes) and mitochondrial membranes during Parkin-mediated mitophagy. This model underscores the importance of liquid-liquid phase separation in facilitating membrane-membrane contacts, likely through the generation of capillary forces.

**Keywords** Autophagy; Mitophagy; Liquid–Liquid Phase Separation; Optineurin; Wetting
**Subject Category** Autophagy & Cell Death

## Introduction

Macroautophagy (hereafter referred to as autophagy) is an intracellular degradation process mediated by a double-membraned organelle termed the autophagosome (Nakatogawa, 2020; Mizushima and Levine, 2020). Cytosolic components such as proteins and organelles are segregated into autophagosomes, which subsequently fuse with lysosomes for degradation. Autophagy can be either non-selective or selective (Lamark and Johansen, 2021; Vargas et al, 2023). Mitophagy, which is a selective type of autophagy, specifically degrades mitochondria and can be induced by mitochondrial damage or cellular stresses (Onishi et al, 2021; Ganley and Simonsen, 2022). For example, upon mitochondrial damage, numerous mitochondrial outer-membrane proteins are ubiquitinated by the PINK1-Parkin system, initiating the recruitment of autophagy adaptors, including optineurin (OPTN), calcium binding and coiled-coil domain 2 (CALCOCO2/NDP52), sequestosome 1 (SQSTM1/p62), Next to BRCA1 gene 1 protein (NBR1), and Tax1 binding protein 1 (TAX1BP1) (Lamark and Johansen, 2021; Vargas et al, 2023). These autophagy adaptors bridge the mitochondrial and autophagosomal membranes by interacting with mitochondrial ubiquitinated proteins via a ubiquitin-binding domain (UBD) and with autophagosomal ATG8 family proteins (i.e. LC3 and GABARAP family proteins in mammals) via the LC3-interacting region (LIR). In addition to their role in supporting efficient engulfment by autophagosomes, OTPN and NDP52 also play crucial roles in inducing mitophagy by recruiting upstream autophagy factors (Lazarou et al, 2015); OPTN recruits ATG9 vesicles and TBK1, which subsequently recruits the PI3K complex, whereas NDP52 recruits the ULK1 complex through interaction with FIP200 (Yamano et al, 2020, 2024; Nguyen et al, 2023; Vargas et al, 2023).

Liquid–liquid phase separation (LLPS) has been described in various biological processes (Musacchio, 2022; Hirose et al, 2023). LLPS involves the formation of liquid-like condensates through dynamic, multivalent interactions between various molecules that often possess intrinsically disordered regions. Emerging evidence suggests connections between autophagy and LLPS at multiple steps, including autophagosome formation and degradation (Noda et al, 2020; Ma et al, 2023). Notably, autophagy adaptors, such as p62, undergo LLPS through multivalent interaction with poly-ubiquitins and form cytosolic condensates (Zaffagnini et al, 2018; Sun et al, 2018). These condensates can be degraded by autophagy, which is termed "fluidophagy" (Agudo-Canalejo et al, 2021). Furthermore, recent studies showed that p62 undergoes LLPS on

[1]Department of Biochemistry and Molecular Biology, Graduate School of Medicine, The University of Tokyo, Tokyo, Japan. [2]Department of Biosystems Science, Institute for Life and Medical Sciences, Kyoto University, Sakyo-ku, Kyoto, Japan. [3]Interdisciplinary Theoretical and Mathematical Sciences (iTHEMS) Program, RIKEN, Wako, Saitama, Japan. [4]Institute of Biology, Humboldt-Universität zu Berlin, Berlin, Germany. [5]International Research Center for Neurointelligence (WPI-IRCN), UTIAS, The University of Tokyo, Tokyo, Japan. [6]Present address: Leibniz Forschungsinstitut für Molekulare Pharmakologie (FMP), Berlin, Germany. [7]Present address: School of Science/Graduate School of Nanobioscience, Yokohama City University, Yokohama, Japan. [8]Present address: Department of Cell Biology, Harvard Medical school, Boston, MA, USA. [9]Present address: University of Cologne, Faculty of Medicine and University Hospital Cologne, Cologne, Germany. ✉E-mail: nmizu@m.u-tokyo.ac.jp

mitochondria and lysosomes that are subjected to selective autophagy (Peng et al, 2021; Gallagher and Holzbaur, 2023).

Generally, phase-separated condensates can deform when they contact a rigid surface in order to minimise the overall energy of the system, a phenomenon known as "wetting" (Kusumaatmaja et al, 2021b; Gouveia et al, 2022). The wetting behaviour of intracellular condensates is determined by the interfacial tensions between the condensate–membrane, condensate–cytosol, and cytosol-membrane interfaces. Depending on the relative strengths of these interfacial tensions, the condensates exhibit partial wetting, complete wetting, or de-wetting. In contrast, elastic surfaces, including membranes to which phase-separated condensates adhere, also deform (Kusumaatmaja et al, 2021a, 2021b; Gouveia et al, 2022; Mangiarotti et al, 2023). This indeed happens during fluidophagy; autophagosomal membranes bend along the surface of p62 condensates (Agudo-Canalejo et al, 2021). Deformation of a membrane by a phase-separated condensate is determined by the energy balance between membrane deformation and the surface tension of the condensate, which is known as elastocapillarity (Style et al, 2017; Kusumaatmaja et al, 2021b; Mangiarotti et al, 2023).

By forming capillary bridges, wetting droplets can also induce adhesion of surfaces (Wexler et al, 2014). Whether LLPS and capillary bridges also contribute to the adhesion of cellular membranes is not presently known. Therefore, we hypothesised that phase-separated condensates of autophagy adaptors can wet both mitochondria and autophagic membranes, including ATG9 vesicles and isolation membranes (also called phagophores), thereby promoting their contact during Parkin-mediated mitophagy. In this study, we employ live-cell imaging and a mathematical model to provide evidence that the autophagy adaptors form phase-separated condensates on the surface of the mitochondrial membrane, resulting in sheet-like condensates that cover the surface of ubiquitinated mitochondria upon mitophagy induction. These condensates accumulate between mitochondria or between mitochondria and autophagic membranes, exhibiting a dynamic nature with a liquid-like property. Furthermore, we demonstrate the essential role of these dynamic condensates in the recruitment of ATG9 vesicles and the initiation of mitophagy. Based on these results, we propose a dynamic liquid-like model, rather than a stoichiometric model, to describe the roles of autophagy adaptors in Parkin-mediated mitophagy.

# Results

## Autophagy adaptors show distinct distributions during Parkin-mediated mitophagy

To better understand the role of each autophagy adaptor, we analyzed their localisation during Parkin-mediated mitophagy. When mitophagy was induced by treatment with the mitochondrial uncoupler carbonyl cyanide *m*-chlorophenylhydrazone (CCCP) in HeLa cells expressing exogenous Parkin, the autophagy adaptors p62, OPTN, NBR1, NDP52 and TAX1BP1 translocated to mitochondria as previously observed (Fig. 1A,B) (Geisler et al, 2010; Wong and Holzbaur, 2014; Heo et al, 2015; Moore and Holzbaur, 2016; Gallagher and Holzbaur, 2023). Ubiquitin and all of these autophagy adaptors were distributed evenly on the surface of separate mitochondria (Mt; Fig. 1A). In contrast, these adaptors

exhibited distinct localisation patterns during the formation of isolation membranes on mitochondria (Mt–IM; Fig. 1B,C). The signals of OPTN and NDP52 were enriched in areas where LC3B signals colocalized, while they were mostly absent from the LC3B-negative side of the same mitochondria (Fig. 1B,C). Endogenous OPTN and NDP52 also showed clear enrichment on the LC3B-positive areas in comparison to the LC3B-negative side, confirming that this localisation is not caused by the overexpression of adaptors (Fig. 1D). However, this inhomogeneous distribution pattern was not observed for ubiquitin, p62, NBR1, and TAX1BP1 (Fig. 1B,C). Unlike OPTN and NDP52, p62 showed enrichment between clustered mitochondria (Mt–Mt), which is consistent with previous reports (Fig. EV1A,B) (Wong and Holzbaur, 2014), which appears to be consistent with the role of p62 in promoting mitochondrial clustering (Narendra et al, 2010; Okatsu et al, 2010). These accumulation patterns cannot be explained by the stoichiometric interaction of the autophagy adaptors with ubiquitinated proteins that are evenly distributed on mitochondria.

The localisation of autophagy adaptors may be affected by their interaction with other adaptors (Turco et al, 2021; Gubas and Dikic, 2022). To examine the localisation of each adaptor on its own, we used HeLa cells lacking all five autophagy adaptors (penta KO cells) to exclude the effect of other adaptors (Lazarou et al, 2015). In penta KO cells, OPTN and NDP52 exhibited significant enrichment in LC3B-positive areas, similar to that observed in wild-type cells, suggesting that this accumulation occurred independently of heterologous interaction with other adaptors (Fig. 1E,F). Mitophagy was not restored by exogenous expression of p62 or NBR1 in penta KO cells, and thus colocalization with LC3 could not be tested for these two adaptors (Lazarou et al, 2015). The accumulation of p62 between clustered mitochondria was also observed in penta KO cells (Fig. EV1C,D). Given the uniform distribution of ubiquitinated proteins on mitochondrial surfaces, these data suggest that autophagy adaptors exhibit non-stoichiometric enrichment between membranes. Hereafter, we used penta KO cells to study each autophagy adaptor individually.

## OPTN and NDP52 show a dynamic exchange between the mitochondrial surface and cytosol

We hypothesised that the non-stoichiometric distribution of the autophagy adaptors on ubiquitinated mitochondria could be explained by LLPS. Typical condensates generated by LLPS exhibit rapid exchange of their components, which is often demonstrated by fluorescence recovery after photobleaching (FRAP) analysis (Taylor et al, 2019; McSwiggen et al, 2019). We first conducted FRAP experiments for separate (i.e. unclustered) mitochondria in CCCP-treated cells. As expected, ubiquitin showed virtually no recovery because it was conjugated to mitochondrial membrane proteins (Fig. 2A,B). Additionally, p62 recovered only slightly, suggesting a minute exchange between the mitochondrial surface and cytosol (Fig. 2A,B). In contrast, OPTN and NDP52 showed a rapid recovery, indicating a dynamic exchange of these adaptors between the mitochondrial surface and the cytosol (Fig. 2A,B).

Next, we examined the dynamics of adaptors between mitochondria and isolation membranes. OPTN and NDP52 showed partial recovery (Fig. 2C,D). Although this result indicates some exchange of these adaptors between membranes, the incomplete recovery suggests the existence of a gel-like or immobile

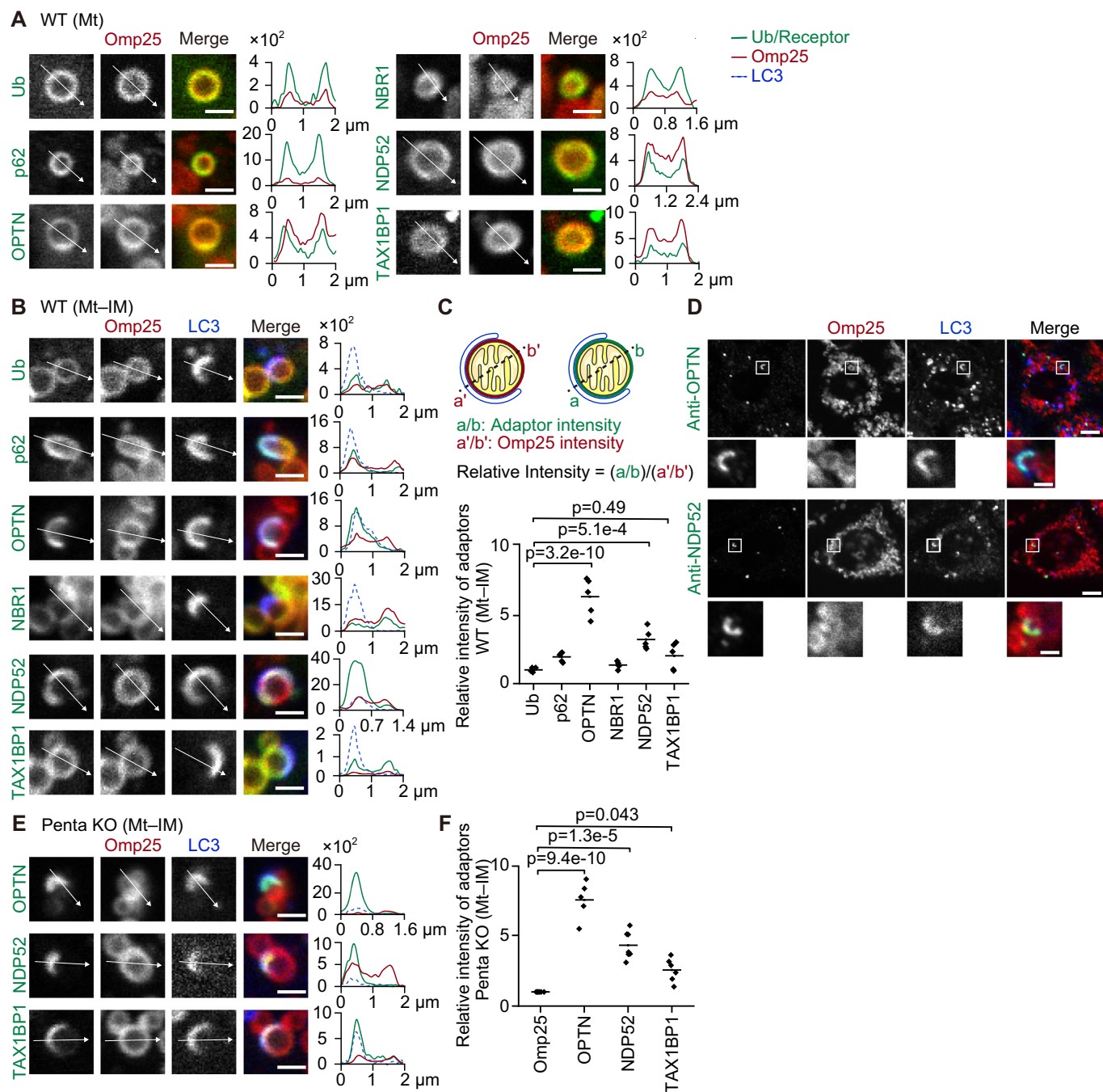

fraction. Treatment with 1,6-hexanediol, which can be used to dissolve condensates, dispersed OPTN and NDP52, but not p62, from mitochondria without isolation membranes (Figs. 2E and EV2A). In contrast, 2,5-hexanediol, which is considered by some studies as a negative control for 1,6-hexanediol, did not disperse the OPTN or NDP52 condensates (Fig. EV2B). These data suggest that OPTN and NDP52 accumulation is supported by weak interactions. In contrast, OPTN and NDP52 at the mitochondria-isolation membrane contact sites remained unaffected by 1,6-hexanediol (Fig. EV2C). This may be due to the additional interaction with ATG8 proteins, not only with ubiquitin. These results indicate that

OPTN and NDP52 on ubiquitinated mitochondria are mobile and can quickly exchange between the mitochondrial surface and the cytosol, supporting the hypothesis that they form phase-separated condensates on damaged mitochondria.

## Mathematical models of condensate formation and localisation of autophagy adaptors

To simulate the behaviour of autophagy adaptors in the context of phase-separated condensates, we developed mathematical models (see Materials and Methods for details). First, we considered the

**Figure 1.    Autophagy adaptors show distinct distributions during Parkin-mediated mitophagy.**

(A, B) Representative images (left) and spline graphs of the intensity profiles along the indicated arrows (from position 0) in the respective images (right) of wild-type HeLa cells expressing one of the GFP-tagged autophagy adaptors or ubiquitin, mRuby–Omp25 and HaloTag7–LC3B at 45 min after CCCP treatment. Separate mitochondria without contacting other mitochondria or isolation membranes (A; Mt) or mitochondria with an isolation membrane (B; Mt–IM) are shown. The y-axis in each of the graphs indicates the fluorescence intensity. Scale bars indicate 1 μm. (C) The relative intensity of GFP-tagged adaptors or ubiquitin (a and b) in LC3B-positive areas compared with LC3B-negative areas in panel B. GFP signals were normalised to the mRuby–Omp25 signals (a' and b') using the formula, relative intensity = (a/b)/(a'/b'), where a and a' represent the intensities in the LC3B-positive area, and b and b' represent the intensities in the LC3B-negative area. Solid horizontal bars indicate the means, and dots indicate the data from five structures. Differences were statistically analyzed by one-way analysis of variance with Dunnett's post-hoc test. (D) Immunostaining of endogenous OPTN or NDP52 in wild-type HeLa cells expressing mRuby–Omp25 and HaloTag7–LC3B at 45 min after CCCP treatment. Scale bars indicate 4 μm and 1 μm (magnified images). (E) Representative images (left) and spline graphs of the intensity profiles along the indicated arrows (from position 0) in the respective images (right) of HeLa cells lacking all five autophagy adaptors (penta KO cells) expressing indicated GFP-tagged autophagy adaptors, mRuby–Omp25, and HaloTag7–LC3B at 45 min after CCCP treatment. Mitochondria with an isolation membrane (Mt–IM) are shown. The y-axis in each of the graphs indicates the fluorescence intensity. Scale bars indicate 1 μm. (F) The relative intensity of GFP-tagged adaptors or ubiquitin (a and b) in LC3B-positive areas compared with LC3B-negative areas in panel (E). Relative intensities were calculated as in (C). Solid horizontal bars indicate the means, and dots indicate the data from five structures. Differences were statistically analyzed by one-way analysis of variance with Dunnett's post-hoc test. Source data are available online for this figure.

case of two mitochondria approaching each other (Fig. 3A). In Parkin-mediated mitophagy, various proteins in the mitochondrial outer membrane are ubiquitinated. We modelled the mitochondrial outer membrane as a circle with a diameter of $D_{Mt} = 600$ nm. On these mitochondria, we assumed a ring-shaped ubiquitin layer with a thickness $w_{Ub} = 20$ nm and area-fraction (hereafter referred to as concentration) $\phi_{Ub} = 0.1$ (Milo and Phillips, 2015; Berry et al, 2018) (Fig. 3A). The closest distance between the two mitochondrial outer membranes was set to $d_{Mt}$ (Fig. 3A). Here, we considered the dynamics of concentration changes of the adaptor proteins and solvent components. We assumed that mitochondria were very large compared with proteins and did not move on a similarly short time scale, and that ubiquitin was tightly bound to mitochondrial membranes and immobile, as suggested by the FRAP experiment (Fig. 2B). The adaptor proteins were assumed to diffuse freely in solution and to bind to ubiquitin and themselves (self-interactions) with strengths $\chi_{Ub}$ and $\chi_{self}$, respectively. The adaptors showed accumulation and distribution on mitochondria similarly to what was observed in cells (Fig. EV1) when $\chi_{Ub} \geq 12k_BT$ and $\chi_{self} = 3k_BT$ (Fig. EV3A), with the Boltzmann constant $k_B$ and temperature $T$. Therefore, we used $\chi_{Ub} = 12k_BT$ and $\chi_{self} = 3k_BT$ for further analyses.

Using the mathematical model, we assessed how the condensates deformed when the distance between two mitochondria $d_{Mt}$ was changed. When two mitochondria were far apart ($d_{Mt} = 900$ nm), the model predicted that adaptors cover the entire mitochondrial surface (Fig. 3B). When the mitochondria were close enough to come into contact ($d_{Mt} = 60$ nm), the adaptors moved to the area between the mitochondria and filled the cleft between them (arrowheads, Fig. 3B).

Next, we considered the case in which a growing isolation membrane surrounded and enclosed a single mitochondrion (Fig. 3C,D). A mitochondrion with ubiquitin was modelled as mentioned above, and the isolation membrane was modeled as a region of thickness $w_{IM} = 50$ nm bound by two semicircles with the same centre as the mitochondrion. The closest distance between the isolation membrane and the mitochondrial outer membrane was $d_{IM} = 100$ nm, and the isolation membrane bent at an angle $\alpha$ to surround the mitochondrion. The surface of the isolation membrane was assumed to be covered by an LC3 region with a thickness $w_{LC3} = 20$ nm and concentration $\phi_{LC3} = 0.1$. The adaptors interacted with both LC3 and ubiquitin with strengths $\chi_{LC3}$ and $\chi_{Ub}$, respectively. A clear accumulation of the autophagy adaptor in the region in contact with the isolation membrane, along

with depletion from the non-contacting region, was observed at $\chi_{LC3} \geq 12k_BT$ (Fig. EV3B), similar to the OPTN distribution observed in cells (Fig. 1B). Therefore, we used $\chi_{LC3} = 12k_BT$ for further analyses.

Using this model, we determined the distribution of the adaptors when the isolation membrane appeared and elongated (changing the angle $\alpha$ surrounding the mitochondrion). In the absence of the isolation membrane, the adaptors wet the entire mitochondrial surface uniformly (Fig. 3D). However, with the appearance of the isolation membrane, the adaptors remobilized to the area between the isolation membrane and mitochondrion, and this adaptor-enriched region elongated together with the isolation membrane. These modelling results suggest that the distribution of the adaptors on a mitochondrion can change upon contact with another mitochondrion or an isolation membrane.

## OPTN condensates redistribute upon membrane contact

Our mathematical models predict that the OPTN distribution changes when mitochondria cluster or are engulfed by isolation membranes. We then validated these predictions by using live-cell imaging. When two small mitochondria with uniform OPTN distributions on their surface approached each other, OPTN redistributed and formed a smooth surface that covered the two adjacent mitochondria (Fig. 4A–D; Movie EV1). Notably, strong OPTN enrichment was observed in the cleft between the two mitochondria (arrowheads, Fig. 4A,C), showing a pattern distinct from that of the outer membrane protein Omp25 (OPTN signal peaks localised outside of Omp25 signal peaks) (Fig. 4B,D). This phenomenon was apparent when the size of both (Fig. 4A) or one (Fig. 4C; Movie EV1) of the two mitochondria was less than 1 μm. Although OPTN formed sheet-like rather than spherical condensates, this phenomenon appeared to be similar to condensate coalescence, which is one of the hallmarks of liquid-like condensates (Hyman et al, 2014).

Furthermore, when an isolation membrane started to engulf a mitochondrion, OPTN, which was initially distributed uniformly on the mitochondrial surface, underwent redistribution to the area contacting the isolation membrane (Fig. 4E; Movie EV2). OPTN signals on the isolation membrane-negative side of the mitochondrion diminished during this process (Fig. 4E,F). The OPTN-enriched region expanded together with the isolation membrane thereafter (Fig. 4E). These live-cell observations are consistent with

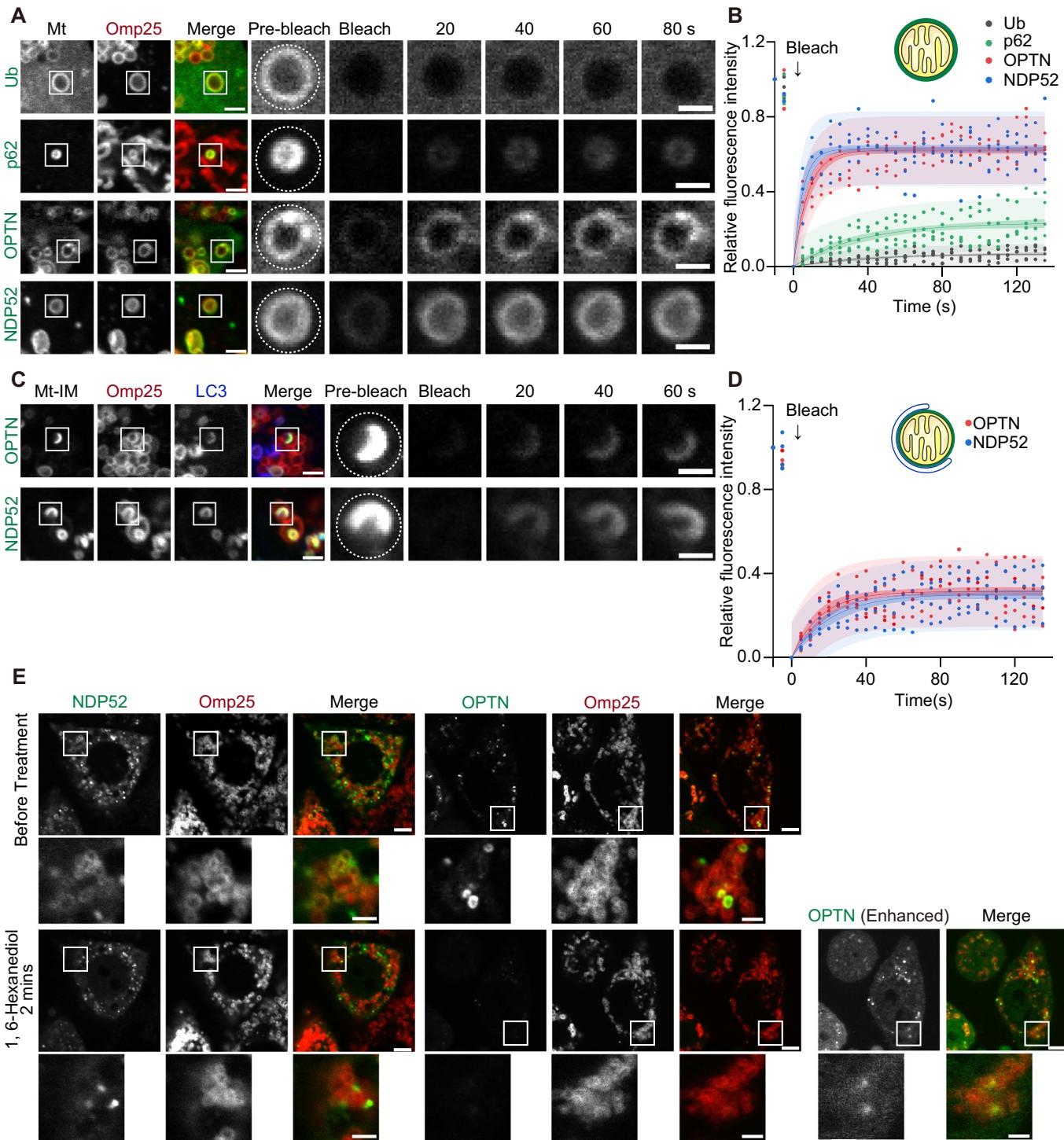

**Figure 2. OPTN and NDP52 show a dynamic exchange between the mitochondrial surface and cytosol.**

(A–D) Representative images (A, C) and quantification (B, D) of GFP fluorescence recovery after photobleaching (FRAP) on separate mitochondria (Mt) (A, B) or on mitochondria with an isolation membrane (Mt–IM) (C, D) in HeLa cells lacking all five autophagy adaptors (penta KO cells) expressing GFP-tagged adaptors or ubiquitin at 45 min after CCCP treatment. The photobleached areas are indicated by dotted lines. The magnified panels display time-lapse images of the photobleached areas. Scale bars indicate 2 and 1 μm (magnified images). Data were collected from four structures and were fitted to the equation $y = a^{*}(1 - \exp(-b^{*}x))$. The dark-shaded areas represent the 95% confidence intervals, and the light-shaded areas represent the 95% prediction intervals. (E) Penta KO cells expressing GFP-OPTN and GFP-NDP52 at 45 min after treatment with CCCP and wortmannin. Wortmannin was added to inhibit autophagosome formation so that OPTN is not sequestered in a closed compartment. Images of cells before (upper panels) and 2 min after (lower panels) the addition of 10% 1,6-hexanediol are displayed. Scale bars, 5 and 2 μm (magnified images). Source data are available online for this figure.

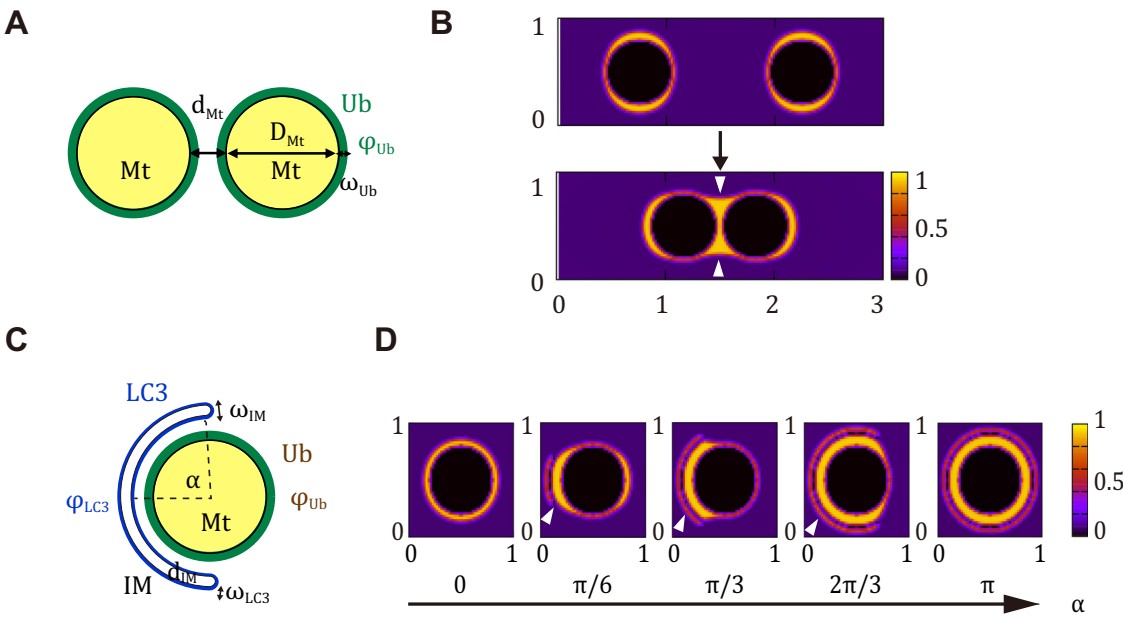

**Figure 3. Mathematical models of condensate formation and localisation of autophagy adaptors.**

(A) Schematic diagram of two separated mitochondria. (B) Sheet-like condensates of autophagy adaptors on mitochondria deform when the inter-mitochondrial distance $d_{Mt}$ was changed. The strength of adaptor self-interaction ($\chi_{self}$), adaptor–ubiquitin interaction ($\chi_{Ub}$), mitochondrial area exclusion ($\chi_{Mt}$) and surface tension ($\sigma$) were set to $\chi_{self} = 3k_BT$, $\chi_{Ub} = 12k_BT$, $\chi_{Mt} = 5k_BT$, and $\sigma = k_BT$, respectively. The colour indicates the adaptor concentration ($\phi_{ad}$). Arrowheads indicate autophagy adaptors filling the cleft between the two mitochondria. (C) Schematic diagram of a single mitochondrion with an isolation membrane. (D) Sheet-like condensates of autophagy adaptors on mitochondria deformed when the isolation membrane (arrowheads) appeared, and the angle $\alpha$ surrounding the mitochondria changed. The strength of adaptor self-interaction ($\chi_{self}$), adaptor–ubiquitin interaction ($\chi_{Ub}$), adaptor-LC3 interaction ($\chi_{LC3}$), mitochondrial area exclusion ($\chi_{Mt}$), isolation membrane area exclusion ($\chi_{IM}$), and surface tension ($\sigma$) were set to $\chi_{self} = 3k_BT$, $\chi_{Ub} = \chi_{LC3} = 12k_BT$, $\chi_{Mt} = \chi_{IM} = 5k_BT$, and $\sigma = k_BT$, respectively. The colour indicates the adaptor concentration ($\phi_{ad}$).

the predictions of our mathematical models (Fig. 3). Taken together, our experimental data support the hypothesis that OPTN forms sheet-like phase-separated condensates on the surface of ubiquitinated mitochondria, exhibiting liquid-like properties.

## Liquid-like property of OPTN condensates is required for mitophagy

To investigate the importance of the liquid-like nature of OPTN condensates on mitochondria, we sought to reduce their fluidity by strengthening the interaction between ubiquitin and GFP–OPTN constructs. To achieve this, we utilised the anti-green fluorescent protein (GFP) nanobody, which exhibits a high binding affinity to GFP (Rothbauer et al, 2008) (Fig. 5A). Consequently, the interaction between anti-GFP nanobody-fused ubiquitin and GFP–OPTN was stronger than that between ubiquitin and GFP–OPTN (Fig. 5B). Expression of anti-GFP nanobody-fused ubiquitin (nanobody–mRuby–Ub) did not affect the recruitment of GFP–OPTN to mitochondria (Fig. 5A, (i) and (ii)) but partially reduced its FRAP recovery (Fig. 5C), suggesting a reduction in mobility. We then evaluated the effect of reduced OPTN fluidity on mitophagy using the recently developed HaloTag cleavage assay (Yim et al, 2022). This assay utilises the ligand-dependent conformational change of HaloTag (Halo); ligand-free Halo is efficiently degraded in lysosomes, whereas ligand-bound Halo becomes resistant to lysosomal degradation. When Halo is

expressed in the mitochondrial matrix by fusing to the mitochondrial presequence of Fo-ATPase subunit 9 and SNAP-tag (mtHalo–SNAP), we can measure the extent of mitochondrial degradation by quantifying the amount of processed free Halo out of the total amount of Halo (mtHalo–SNAP + processed Halo). Mitophagy activity, which was impaired in penta KO cells, was restored by the expression of GFP–OPTN (Fig. 5D,E). Mitophagy activity was partially reduced in cells expressing GFP–OPTN and nanobody–mRuby–Ub (Fig. 5D,E).

The partial reduction in both FRAP recovery and mitophagy activity may be attributed to the remaining interaction between OPTN and endogenous ubiquitin. Therefore, we deleted the ubiquitin-binding domain in ABINs and NEMO (UBAN domain) from OPTN (GFP–OPTNΔUBAN), abolishing its interaction with ubiquitinated proteins (Fig. 5B). GFP–OPTNΔUBAN failed to accumulate on mitochondria in the presence of mRuby–Ub (Fig. 5A, (iii)). In contrast, the expression of nanobody–mRuby–Ub rescued the recruitment of GFP–OPTNΔUBAN to mitochondria (Fig. 5A, (iv)), albeit to a reduced extent, likely reflecting the loss of condensate formation. FRAP experiments revealed that, compared with wild-type GFP–OPTN (Fig. 5A, (i)), GFP–OPTNΔUBAN (Fig. 5A, (iv)) showed a substantial decrease in recovery on mitochondria when co-expressed with nanobody–mRuby–Ub, indicating a reduction in its dynamic properties (Fig. 5C). GFP–OPTNΔUBAN failed to restore mitophagy, even when it was co-expressed with nanobody–mRuby–Ub to rescue ubiquitin binding and mitochondrial localisation (Fig. 5A,B,D,E). This

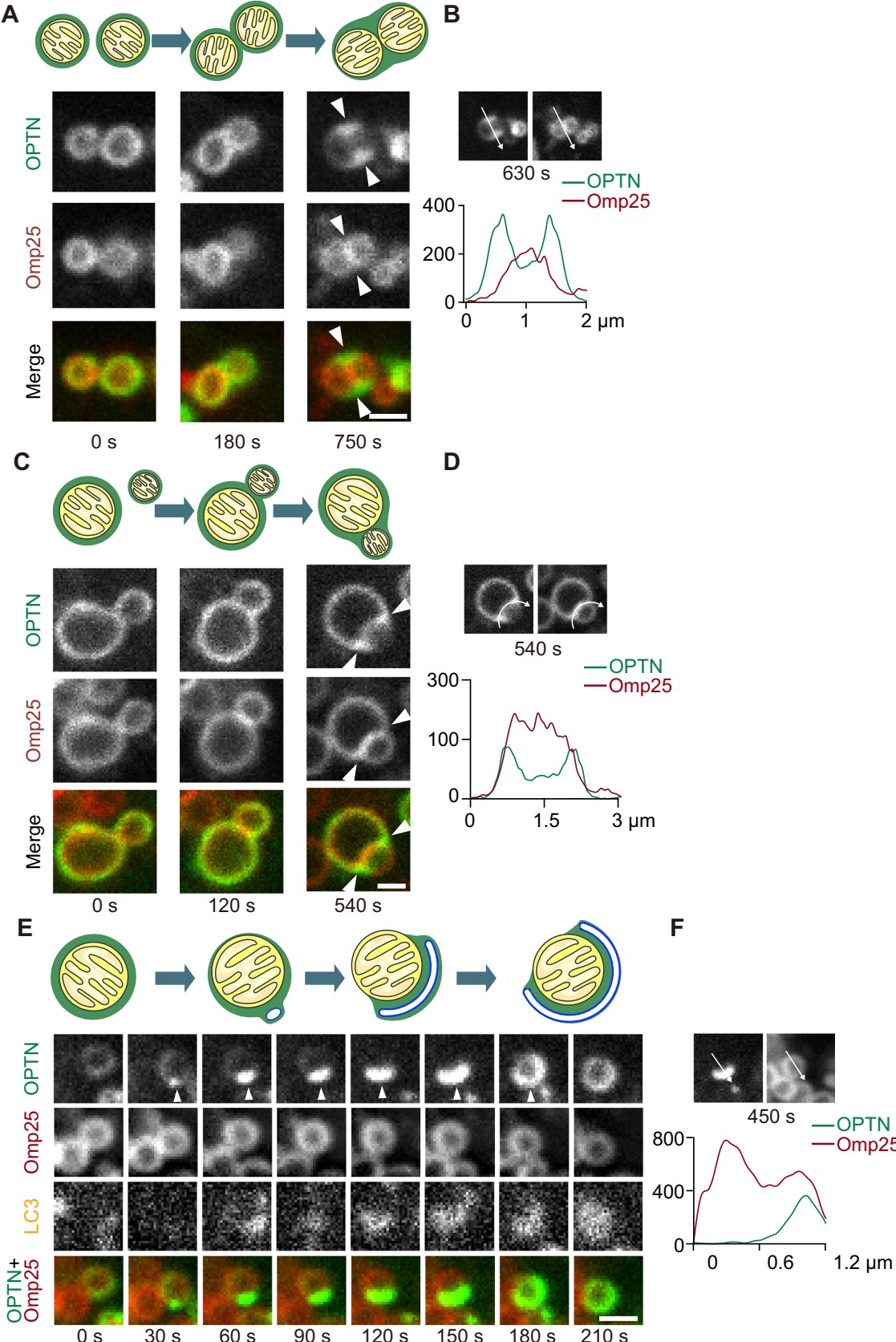

**Figure 4. OPTN condensates redistribute upon contact with membranes.**

(A–D) Time-lapse images (A, C) and spline graphs (B, D) of two separate mitochondria approaching and contacting each other. Events of the attachment of two mitochondria with comparable sizes (A, B) and with differing sizes (C, D) were observed. The line graphs represent the intensity profiles along the indicated arrows (from position 0) in the respective images shown (B, D). The arrowheads indicate redistributed OPTN signals filling the cleft between two adjacent mitochondria. Scale bars indicate 1 μm. See also Movie EV1 for the time-lapse video corresponding to (C). (E, F) Time-lapse images (E) and a spline graph (F) of a mitochondrion with an isolation membrane elongating on its surface. The arrowhead indicates the elongating isolation membrane. The line graph represents the intensity profiles along the indicated arrows (from position 0) in the respective images shown. Scale bars indicate 1 μm. See also Movie EV2 for the time-lapse video corresponding to (E). Source data are available online for this figure.

impairment was indeed due to the loss of binding to ubiquitin because a point mutation that abolished ubiquitin binding (E478G) (Li et al, 2018) also showed normal mitochondrial localisation, low FRAP recovery, and mitophagy defect (Figs. 5D,E and EV4A,B). Furthermore, the impaired mitophagy was not simply due to the altered binding orientation of OPTN. Replacement of its UBAN domain with GFP (OPTNΔUBAN–GFP), mimicking its binding to ubiquitin at the C-terminus, did not restore FRAP recovery or mitophagy activity, although it also showed clear recruitment to mitochondria in the presence of nanobody–mRuby–Ub (Fig. EV4A–D). The restricted orientation of OPTN due to the stoichiometric interaction between GFP and nanobody, as opposed to the more flexible orientation in liquid-like condensates, may also contribute to the impairment of autophagy.

Consistent with the results of mitophagy activity, the accumulation of the ULK1 complex component FIP200, likely representing isolation membranes, was observed in the presence of mRuby–Ub and GFP–OPTN (Fig. EV5A). In contrast, no clear FIP200 accumulation was observed in cells expressing nanobody–mRuby–Ub and GFP–OPTNΔUBAN, suggesting that mitophagy failed at the initiation step in these cells (Fig. EV5A). Together, these data suggest that the localisation of OPTN on mitochondria and its binding to ubiquitin alone is insufficient for mitophagy, and the fluidity of OPTN is important for effective mitophagy induction.

### Liquid-like property of OPTN condensates is required for initiation of mitophagy by recruiting ATG9 vesicles and activating TBK1

ATG9 vesicles are considered to be one of the sources of autophagosomal membranes (Yamamoto et al, 2012). Upon mitophagy induction, ATG9 vesicles accumulate at the sites of autophagosome formation (Itakura et al, 2012). OPTN plays a crucial role in this process by recruiting ATG9 vesicles to induce mitophagy (Yamano et al, 2020). In general, liquid-like condensates can be involved in clustering small vesicles such as synaptic vesicles, which exemplify complete wetting (Milovanovic and De Camilli, 2017; Sansevrino et al, 2023). Indeed, ATG9 vesicles are incorporated into condensates containing a glaucoma-associated OPTN mutant (O'Loughlin et al, 2020) and synapsin (Park et al, 2023). We, therefore, hypothesised that the liquid-like properties of OPTN condensates on mitochondria facilitate the recruitment of ATG9 vesicles. Consistent with previous reports (Itakura et al, 2012; Yamano et al, 2020), ATG9A was recruited to depolarised mitochondria in cells expressing GFP–OPTN (Fig. 6A,B). However, this ATG9A recruitment was almost completely abolished in cells expressing nanobody–mRuby–Ub and GFP–OPTNΔUBAN (Fig. 6A,B), even though GFP–OPTNΔUBAN still retains the ATG9A-interacting domain and the ability to bind with the ATG9 vesicles (Fig. 6C,D). These data suggest that the fluidity of sheet-like

OPTN condensates on the mitochondrial surface is crucial for recruiting ATG9 vesicles and executing mitophagy.

Furthermore, TBK1 is recruited by OPTN to the mitophagy initiation site to be phosphorylated and activated, driving the feed-forward mechanism of OPTN accumulation and TBK1 activation, which in turn phosphorylates the PI3K complex to initiate mitophagy (Nguyen et al, 2023; Yamano et al, 2024). Consistently, a clear accumulation of phosphorylated TBK1 on mitochondria was observed in the presence of GFP–OPTN and mRuby–Ub (Fig. EV5B). In contrast, only very faint signals of phosphorylated TBK1 were observed in the presence of nanobody–mRuby–Ub and GFP–OPTNΔUBAN (Fig. EV5B). These data suggest that the fluidity of OPTN is also important for the feed-forward mechanism between OPTN and TBK1 for the initiation of mitophagy. Together, we propose that autophagy adaptors phase separate on ubiquitinated mitochondria to amplify the initiation signals and wet the ATG9 vesicles to facilitate efficient mitophagy.

## Discussion

### LLPS by autophagy adaptors during Parkin-mediated mitophagy

In the present study, we discovered that autophagy adaptors accumulate on ubiquitinated mitochondria with non-stoichiometric distribution compared with ubiquitin signals. They displayed characteristics consistent with phase-separated structures, including dynamic exchange with cytosol and redistribution upon coalescence. Notably, we observed the enrichment of OPTN at the isolation membrane-positive region, which expanded along with the elongation of the isolation membrane. These findings suggest that phase-separated adaptors facilitate the effective engulfment of mitochondria by the isolation membrane through the wetting effect and capillary forces (Fig. 6E, right panel). Moreover, the fluidity of OPTN condensates is critical for the localisation of ATG9 vesicles onto ubiquitinated mitochondria, indicating that partial or complete wetting of phase-separated adaptors to ATG9 vesicles may be crucial for their recruitment to and/or retention on mitochondria (Fig. 6E, left panel). Therefore, we propose that LLPS of autophagy adaptors plays a critical role in two distinct steps of mitophagy by producing capillary forces, first, at the initiation step by recruiting and retaining the ATG9 vesicles in the condensates and, second, at the autophagosome elongation step by facilitating the attachment between ubiquitinated mitochondria and the isolation membrane. Additionally, our experimental data suggest that NDP52 also exhibits liquid-like properties. Further studies are needed to determine whether condensate formation is necessary for mitophagy induction by NDP52.

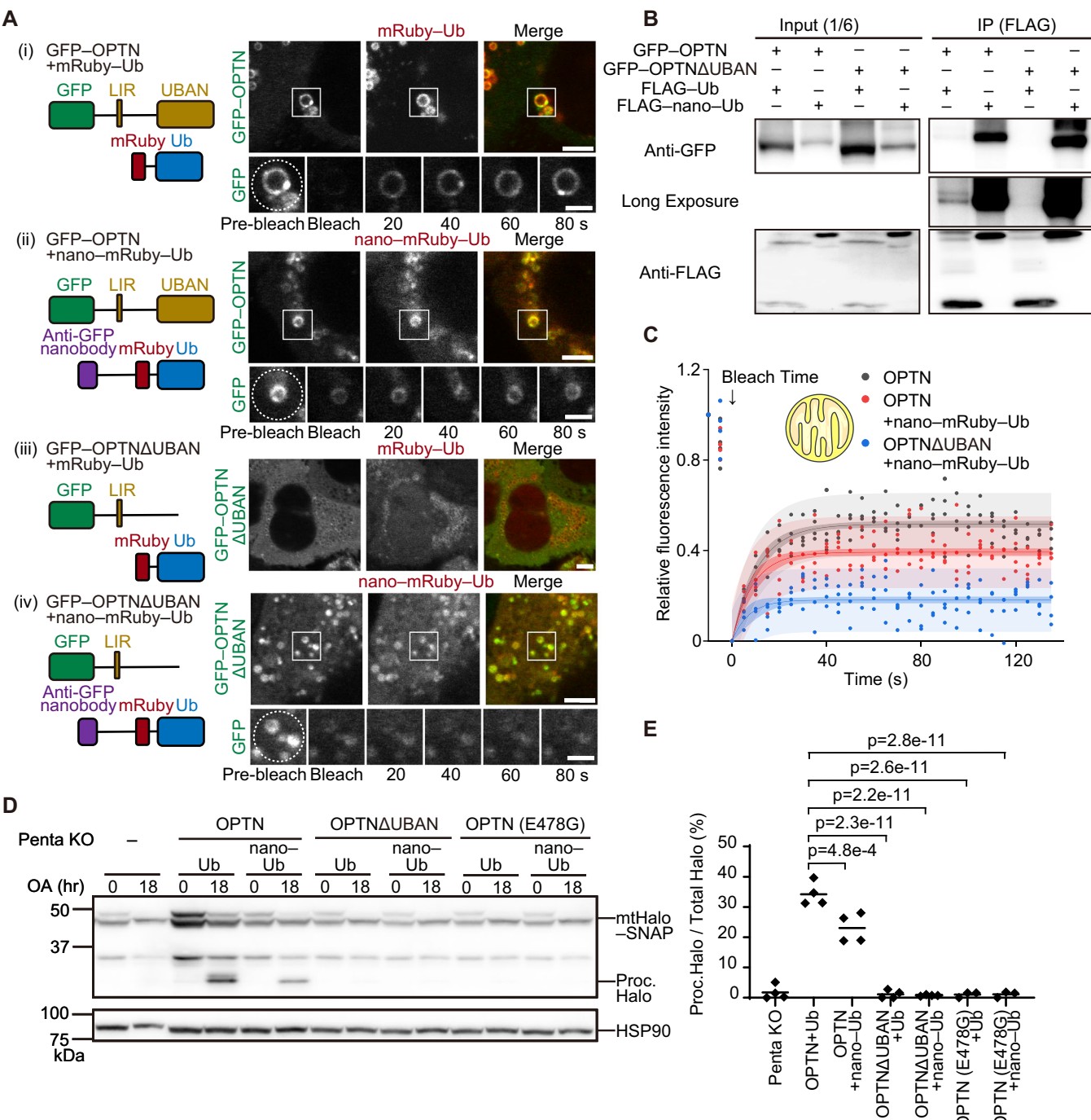

**Figure 5. Liquid-like property of OPTN condensates is required for mitophagy.**

(**A**) HeLa cells lacking all five autophagy adaptors (penta KO cells) expressing both GPF–OPTN and mRuby–Ub (i), both GPF–OPTN and anti-GFP nanobody–mRuby–Ub (ii), boh GFP–OPTNΔUBAN and mRuby–Ub (iii), or both GFP–OPTNΔUBAN and anti-GFP nanobody–mRuby–Ub (iv) at 45 min after CCCP treatment. Time-lapse images of GFP FRAP are shown. Photobleached areas are circled by dotted lines. Scale bars indicate 4 μm and 2 μm (magnified images). (**B**) Interaction between GFP–OPTN or OPTN mutants, and FLAG-Ub or FLAG-nanobody-Ub was investigated by immunoprecipitation with an anti-FLAG antibody and immunoblotting with an anti-GFP antibody. (**C**) Quantification of GFP FRAP on separate mitochondria (Mt) in penta KO cells expressing both GFP–OPTN and mRuby–Ub (i; grey), both GFP–OPTN and anti-GFP nanobody–mRuby–Ub (ii; red), or both GFP–OPTNΔUBAN and anti-GFP nanobody–mRuby–Ub (iv; blue) at 45 min after CCCP treatment. Data were collected from four structures and were fitted to the equation $y = a*(1 - \exp(-b*x))$. The dark shading represents the 95% confidence intervals, and the light shading represents the 95% prediction intervals. (**D, E**) Representative data (**D**) and quantification (**E**) of HaloTag (Halo) processing assay using cells expressing the indicated OPTN and Ub constructs. Cells expressing the mtHalo–SNAP mitophagy reporter were treated without (0 h) and with oligomycin and antimycin for 18 h. The amount of processed Halo (proc. Halo) indicates the relative amount of mitochondria degraded in lysosomes. Solid horizontal bars indicate the means, and dots indicate the data from at least three independent cultures. Differences were statistically analyzed by one-way analysis of variance with Dunnett's post-hoc test. Source data are available online for this figure.

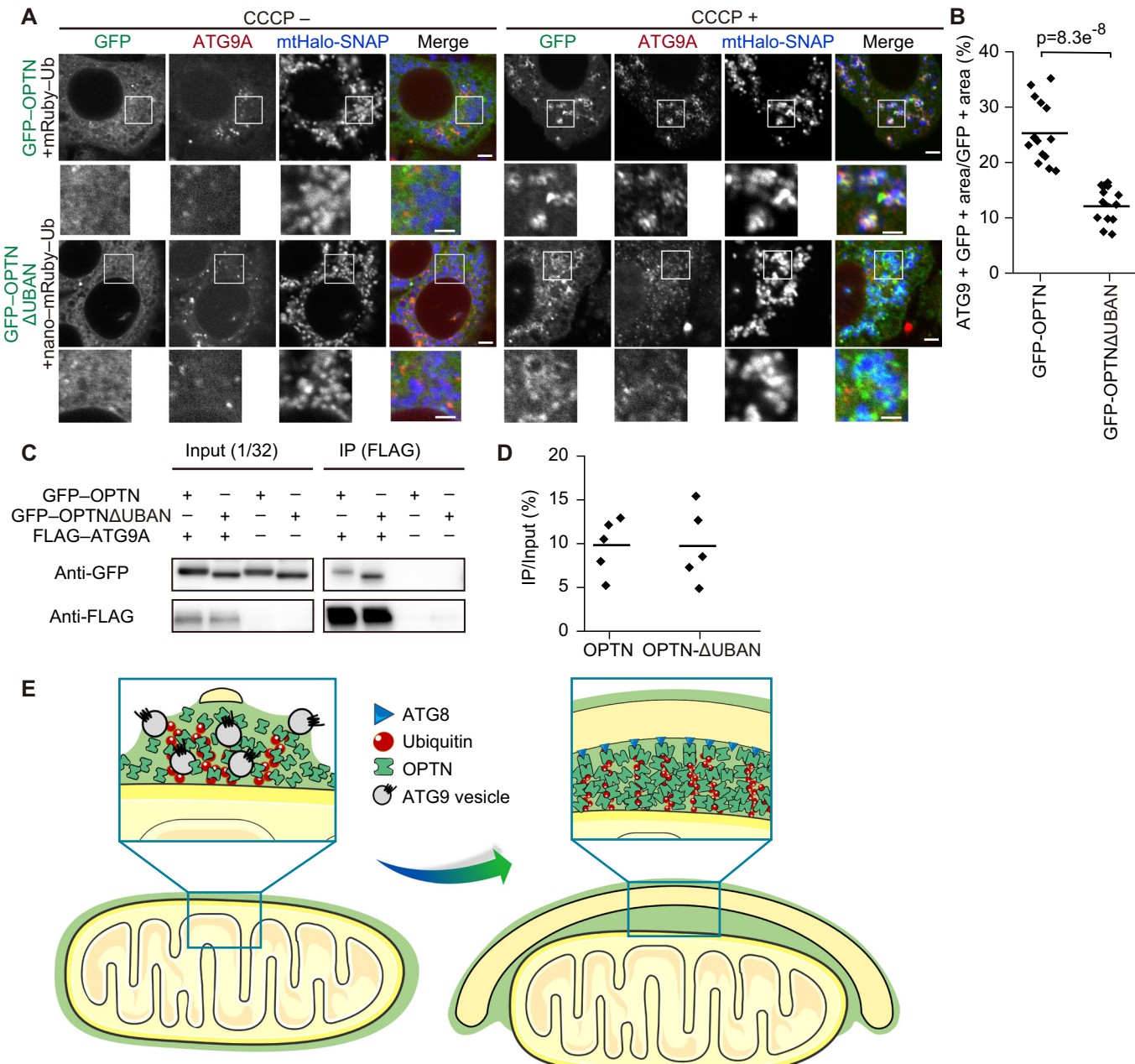

**Figure 6. Liquid-like property of OPTN condensates is required for the recruitment of ATG9 vesicles.**

(A, B) Localisation of ATG9 vesicles under normal (CCCP−) or mitophagy-inducing conditions (CCCP+, 60 min) (A) and quantification (B) of ATG9A colocalization with mitochondria. Endogenous ATG9 was immunostained in penta KO cells expressing both GFP–OPTN and mRuby–Ub or both GFP–OPTNΔUBAN and anti-GFP nanobody–mRuby–Ub (nano-mRuby–Ub) together with mitochondrially targeted Halo-SNAP (mtHalo-SNAP). Scale bars indicate 4 and 2 μm (magnified images). Solid horizontal bars indicate the means, each dot indicates the mean value from one field of view with ≥10 cells ($n = 15$). Differences were statistically analyzed by Welch's two-tailed $t$-test. (C, D) Representative data (C) and quantification (D; $p = 0.99$) of organelle immunoprecipitation in cells expressing GFP–OPTN or GFP–OPTNΔUBAN in the presence or absence of FLAG-ATG9A. ATG9 vesicles were immunoprecipitated by FLAG M2 beads, and GFP was blotted to detect the interaction between GFP–OPTN (WT or ΔUBAN) and ATG9 vesicles at 60 min after CCCP treatment. Solid horizontal bars indicate the means, and dots indicate the data from five independent experiments. Differences were statistically analyzed by Welch's two-tailed $t$-test. (E) The models of ATG9 vesicle recruitment (left) and isolation membrane elongation (right), mediated by liquid-like OPTN condensates. Partial or complete wetting of phase-separated OPTN enables the recruitment of ATG9 vesicles to ubiquitinated mitochondria (left). Wetting of OPTN on mitochondria and isolation membranes also facilitates the engulfment of mitochondria by the isolation membranes (right). Source data are available online for this figure.

## LLPS in bulk and selective autophagy

Recent reports have shed light on the importance of LLPS in selective autophagy (Noda et al, 2020). In lysophagy, p62 undergoes LLPS on lysosomes to facilitate efficient lysophagy together with HSP27, which maintains the fluidity of p62 and prevents its gelation (Gallagher and Holzbaur, 2023). This may also be mediated by the wetting between damaged lysosomes and the isolation membrane by phase-separated p62. Indeed, p62 itself forms phase-separated condensates along with ubiquitinated proteins, producing capillary forces by the wetting effect to facilitate the engulfment by the isolation membrane (Agudo-Canalejo et al, 2021). In our study, p62 exhibited low mobility, as determined by FRAP and 1,6-hexanediol experiments (Figs. 2B and EV2A). While these data may suggest that p62 is immobile, they appear to contradict the observed redistribution of p62 to the area between mitochondria upon clustering (Fig. EV1A,B). Thus, we speculate that p62 may exhibit some degree of fluidity despite its low exchange between the condensate and cytosol. Similarly to p62, Ape1, the cargo of the cytosol-to-vacuole targeting (Cvt) pathway in yeast, forms gel-like condensates with Atg19 on the surface of Ape1 condensates to be sequestered by the isolation membrane (Yamasaki et al, 2020). This association could also be mediated by partial wetting. These examples, together with our observations, highlight the importance of LLPS and the wetting phenomena in various selective autophagy processes.

LLPS may also be important for non-selective bulk autophagy. In yeast, the Atg1 complex (the ULK1 complex in mammals) forms pre-autophagosomal structures (PASs), which have been shown to be condensates driven by LLPS (Fujioka et al, 2020). Phase separation in this context provides a high local concentration of the Atg1 complex to support the autoactivation of Atg1 kinase (Yamamoto et al, 2016). Mammalian autophagy factors also accumulate at the autophagosome formation site; FIP200, a subunit of the autophagy-initiation complex, undergoes LLPS triggered by calcium transients (Zheng et al, 2022). The liquid-like nature of these autophagy-initiating structures may also be important for the recruitment of Atg9/ATG9 vesicles through the wetting effect.

## The importance of phase separation during mitophagy

In contrast to Parkin-mediated mitophagy, hypoxia-induced mitophagy does not involve autophagy adaptors, and instead, the outer mitochondrial membrane proteins NIX (also known as BNIP3L) and BNIP3 directly interact with ATG8 homologues on the isolation membrane (Novak et al, 2010; Onishi et al, 2021). In this context, ATG8 proteins interact with NIX or BNIP3, likely in a one-to-one manner. This raises the question of why phase separation of autophagy adaptors is essential for Parkin-mediated mitophagy. One possibility is that the quick induction and execution of Parkin-mediated mitophagy requires amplification of the reactions through increased local concentration of adaptors and autophagy-related proteins facilitated by LLPS. This is consistent with the fact that mitophagy-inducing signals are also amplified by a mechanism involving phosphorylated ubiquitin (Kane et al, 2014; Kazlauskaite et al, 2014; Koyano et al, 2014) and OPTN-TBK1 axis (Nguyen et al, 2023; Yamano et al, 2024). Indeed, we observed that the amount of OPTN recruited to mitochondria was greater with wild-type OPTN than with OPTNΔUBAN (in the presence of nanobody-Ub), likely reflecting the enhanced recruitment by

condensate formation (the reduced recruitment of OPTNΔUBAN may also be due to a lower expression level of nanobody–mRuby–Ub compared to abundant endogenous ubiquitin, which may further contribute to reduced mitophagy activity) (Figs. 5A and EV4A). Alternatively, Parkin-mediated mitophagy requires a high level of precision and selectivity; mitophagy should be induced only for damaged mitochondria, which is ensured by the recruitment and retention of ATG9 vesicles only on phase-separated adaptor-positive mitochondria. This strict specificity would enable proper quality control of the mitochondria.

# Methods

**Reagents and tools table**

| Reagent/resource | Reference or source | Identifier or catalogue number |
|---|---|---|
| **Experimental models** | | |
| HeLa (*H. sapiens*) | RIKEN | RCB007 |
| HeLa penta-knocknout (penta KO) cell (*H. sapiens*) | Gift from Michael Lazarou | N/A |
| human embryonic kidney (HEK) 293 T cell (*H. sapiens*) | RIKEN | RCB2202 |
| **Recombinant DNA** | | |
| HaloTag7 | Promega | G1891 |
| SNAP-tag | New England BioLabs | N9181S |
| pCG-gag-pol | Dr. Teruhito Yasui (National Institutes of Biomedical Innovation, Health and Nutrition, Japan) | N/A |
| pCG-VSV-G | Dr. Teruhito Yasui (National Institutes of Biomedical Innovation, Health and Nutrition, Japan) | N/A |
| pMRX-IPU-(MCS) | (Kitamura et al, 2003; Saitoh et al, 2003) | N/A |
| pMRX-IBU-(MCS) | (Morita et al, 2018) | N/A |
| pMRX-IZU-HA-Parkin | RDB19764 | AddGene #38248 |
| pMRX-IBU-HaloTag7-hLC3B | (Chino et al, 2019) | N/A |
| pMRX-IPU-muGFP-(MCS) | (Scott et al, 2018) | N/A |
| pMRX-IPU-muGFP-Ubiquitin | This study | N/A |
| pMRX-IPU-muGFP-p62 | This study | N/A |
| pMRX-IPU-muGFP-NDP52 | This study | N/A |
| pMRX-IPU-muGFP-NBR1 | This study | N/A |
| pMRX-IPU-muGFP-OPTN | This study | N/A |
| pMRX-IPU-muGFP-TAX1BP1 | This study | N/A |
| pMRX-IHU-mRuby-(MCS) | (Bajar et al, 2016) | N/A |
| pMRX-IHU-mRuby-ubiquitin | This study | N/A |
| pMRX-IHU-mRuby-nano-ubiquitin | This study | N/A |
| pMRX-IPU-3×FLAG-ATG9A | This study | N/A |
| pMRX-IPU-3×FLAG-ubiquitin | This study | N/A |

| Reagent/resource | Reference or source | Identifier or catalogue number |
|---|---|---|
| pMRX-IPU-3×FLAG-nano-ubiquitin | This study | N/A |
| pMRX-IBU-mtHalo–SNAP | Mizushima Lab | N/A |
| **Antibodies** | | |
| rabbit polyclonal anti-OPTN | Proteintech | 10837-AP |
| rabbit polyclonal anti-ATG9A | MBL | PD042 |
| mouse monoclonal anti-Halo | Promega | G9211 |
| mouse monoclonal anti-HSP90 | BD Transduction Laboratories | 610419 |
| rabbit polyclonal anti-GFP | Thermo Fisher Scientific | A6455 |
| rabbit polyclonal anti-NDP52 | Proteintech | 12229-1-AP |
| rabbit polyclonal anti-RB1CC1 | Proteintech | 17250-1-AP |
| rabbit monoclonal anti-phospho-TBK1 (Ser172) | Cell Signaling Technology | 5483 |
| HRP-conjugated goat polyclonal anti-rabbit IgG | Jackson ImmunoResearch Laboratories | 111-035-144 |
| HRP-conjugated goat polyclonal anti-mouse IgG | Jackson ImmunoResearch Laboratories | 115-035-003 |
| Mouse monoclonal anti-Halo | Promega | G9211 |
| **Oligonucleotides and other sequence-based reagents** | | |
| PCR primers | This study | Table EV1 |
| **Chemicals, Enzymes and other reagents** | | |
| Dulbecco's modified Eagle's medium (DMEM) | Sigma-Aldrich | D6546 |
| Foetal bovine serum (FBS) | Sigma-Aldrich | 173012 |
| L-glutamine | Gibco | 25030-081 |
| Lipofectamine 2000 | Thermo Fisher Scientific | 11668019 |
| Polybrene | Sigma-Aldrich | H9268 |
| Puromycin | Sigma-Aldrich | P8833 |
| Blasticidin S hydrochloride | FUJIFILM Wako Pure Chemical Corporation | 022-18713 |
| G418 disulfate aqueous solution | NACALAI TESQUE | 09380-86 |
| Zeocin | Thermo Fisher Scientific | R25005 |
| Carbonyl cyanide *m*-chlorophenyl hydrazine (CCCP) | Sigma-Aldrich | 555-60-2 |
| Oligomycin | Cabiochem | 495455-10MGCN |
| Antimycin A | Sigma-Aldrich | A8674 |
| 1,6-hexanediol | Sigma-Aldrich | 240117-50 G |
| Wortmannin | Sigma-Aldrich | W1628-1MG |
| Anti-FLAG M2 magnetic beads | Sigma-Aldrich | M8823-1ML |
| SF650-conjugated Halo ligand | GoryoChemical | A308-02 |
| SuperSignal West Pico Chemiluminescent Substrate | Thermo Fisher Scientific | 34579 |

| Reagent/resource | Reference or source | Identifier or catalogue number |
|---|---|---|
| **Software** | | |
| OriginPro 2022 | https://www.originlab.com/ | |
| Fiji | https://fiji.sc/ | |
| **Other** | | |
| Olympus Fluoview FV3000 confocal microscope equipped with a 60× oil-immersion objective lens (1.40 NA, Olympus) | Olympus | |
| Olympus SpinSR10 spinning-disk confocal super-resolution microscope equipped with a Hamamatsu ORCA-Flash 4.0 camera, a UPLAPO OHR 100 × (NA 1.50) lens | Olympus | |

## Cell culture and generation of stable cell lines

HeLa cells, HeLa penta-knockout (penta KO) cells and human embryonic kidney (HEK) 293T cells authenticated by RIKEN were cultured in Dulbecco's modified Eagle's medium (DMEM; D6546; Sigma-Aldrich) supplemented with 10% foetal bovine serum (FBS; 173012; Sigma-Aldrich) and 2 mM L-glutamine (25030-081; Gibco) in a 5% $CO_2$ incubator at 37 °C. For stable expression, retrovirus was produced with HEK293T cells transfected with pMRX-IP-based, pMRX-IB-based, pMRX-IU-based, or pMRX-IN-based retroviral plasmids, pCG-VSV-G, and pCG-gag-pol by using Lipofectamine 2000 (11668019; Thermo Fisher Scientific). After transfection, cells were further incubated at 37 °C for 24 h. The viral supernatant was collected by filtration through a 0.45-µm filter unit (Ultrafree-MC; Millipore) and then used for infection. Cells were plated onto 6-cm dishes 18 h before infection, and the medium was replaced with viral supernatant diluted 1.5-fold with 8 µg/mL polybrene (H9268; Sigma-Aldrich). Two days later, cells were selected in a medium containing 2 µg/mL puromycin (P8833; Sigma-Aldrich), 4 µg/mL blasticidin S hydrochloride (022-18713; FUJIFILM Wako Pure Chemical Corporation), 1.5 mg/mL neomycin or 250 µg/mL zeocin (R25005; Thermo Fisher Scientific).

## Plasmids

The pMRX-IPU, pMRX-IBU and pMRX-INU plasmids were generated by modifying the multi-cloning site of pMRX-IP (Saitoh et al, 2002; Kitamura et al, 2003), pMRX-IB, and pMRX-IN, respectively. DNA fragments encoding ubiquitin, p62 (NM_003900.5), OPTN (NM_001008211.1), NBR1 (NM_001291571.2), NDP52 (NM_001261391.2), TAX1BP1 (NM_001206901.1), ATG9A (NM_001077198.3), LC3A (NM_032514.4) and LC3B (NM_022818.5) were inserted into pMRX-IPU, pMRX-INU, or pMRX-IBU. DNAs encoding the HA epitope, monomeric enhanced GFP with the A206K mutation (mGFP), codon-optimised ultra-stable GFP (muGFP) (Scott et al, 2018), codon-optimised mRuby3 (Bajar et al, 2016), HaloTag7 (G1891; Promega), 3×FLAG, and SNAP-tag (New England BioLabs, N9181S) were used for tagging. The mitochondrial presequence of *Neurospora crassa* Fo-ATPase subunit 9 (residues 1–69) was added to HaloTag7-SNAP to deliver the reporter into the mitochondrial matrix

(mtHalo–SNAP) (Eura, 2003). Truncated OPTN (OPTNΔUBAN, aa 445–502) was prepared by PCR-mediated site-directed mutagenesis. The resulting plasmids were sequenced.

## Antibodies and reagents

The primary antibodies used in this study were rabbit polyclonal anti-OPTN (Proteintech, 10837-AP), rabbit polyclonal anti-ATG9A (MBL, PD042), mouse monoclonal anti-Halo (Promega, G9211), mouse monoclonal anti-HSP90 (BD Transduction Laboratories, 610419), rabbit polyclonal anti-GFP (Thermo Fisher Scientific, A6455), rabbit polyclonal anti-NDP52 (Proteintech, 12229-1-AP), rabbit polyclonal anti-RB1CC1 (Proteintech, 17250-1-AP) and rabbit monoclonal anti-phospho-TBK1 (Ser172) (Cell Signaling Technology, 5483) antibodies. The secondary antibodies used were HRP-conjugated goat polyclonal anti-rabbit IgG (Jackson ImmunoResearch Laboratories, 111-035-144) and HRP-conjugated goat polyclonal anti-mouse IgG (Jackson ImmunoResearch Laboratories, 115-035-003) antibodies. To induce mitophagy, HeLa cells were treated with 20 µM carbonyl cyanide *m*-chlorophenyl hydrazine (CCCP; Sigma-Aldrich) for 45 min, or 10 µM oligomycin (Cabiochem, 495455-10MGCN) and 4 µM antimycin A (Sigma-Aldrich, A8674) for 18 h. After cells had been treated with oligomycin and antimycin A for >6 h, 10 µM Q-VD-OPH (SM Biochemicals, SMPH001) was added to block apoptotic cell death. To inhibit the formation of isolation membranes in the 1,6-hexanediol (Sigma-Aldrich, 240117-50 G) treatment experiment, wortmannin (Sigma-Aldrich, W1628-1MG) was added into the medium.

## Immunoprecipitation and immunoblotting

Cell lysates were prepared in a lysis buffer (50 mM Tris-HCl pH 7.4, 150 mM NaCl, 1 mM EDTA, 1% Triton X-100, EDTA-free protease inhibitor cocktail [19543200; Roche]). After centrifugation at 17,700×*g* for 10 min, the supernatants were subjected to immunoprecipitation using anti-FLAG M2 magnetic beads (M8823-1ML; Sigma-Aldrich). Precipitated immunocomplexes were washed three times with lysis buffer and boiled in sample buffer (46.7 mM Tris-HCl, pH 6.8, 5% glycerol, 1.67% sodium dodecyl sulfate, 1.55% dithiothreitol, and 0.02% bromophenol blue). For immunoprecipitation of ATG9 vesicles, cells were disrupted by Dounce homogenisation with hypotonic lysis buffer (10 mM HEPES, pH 7.9, 1.5 mM $MgCl_2$ and 10 mM KCl, EDTA-free protease inhibitor cocktail [19543200; Roche]). Disruption was carried out by applying 35 strokes while the cell suspension was cooled on ice. The supernatants were subjected to immunoprecipitation using anti-FLAG M2 magnetic beads (M8823-1ML; Sigma-Aldrich). For immunoblotting, the samples were separated by SDS-PAGE and transferred to Immobilon-P polyvinylidene difluoride membranes (Millipore, WBKLS0500) with the Trans-Blot Turbo Transfer System (Bio-Rad). After incubation with the relevant antibody in 5% skim milk in 20 mM Tris-HCl, 150 mM NaCl, and 0.1% Tween 20 (02194841-CF; MP Biomedicals), the signals from incubation with SuperSignal West Pico Chemiluminescent Substrate (Thermo Fisher Scientific, 34579) were detected with the FUSION SOLO.7S.EDGE imaging system (Vilber-Lourmat). Contrast and brightness adjustment and quantification were performed using the image processing software Fiji (Schindelin et al, 2012).

## Fluorescence recovery after photobleaching

In-cell fluorescence recovery after photobleaching (FRAP) analyses were performed with an Olympus Fluoview FV3000 confocal microscope equipped with a 60× oil-immersion objective lens (1.40 NA, Olympus). The chamber was maintained at 37 °C and continuously supplied with humidified 5% $CO_2$. Bleaching was performed using 80% laser power (488 or 561 nm laser), and images were captured every 5 s for 30 frames. Recovery curves and fitting were analyzed using OriginPro 2022. The fluorescence intensity of the bleached spot, an unbleached control spot, and the background was measured using the software package Fiji. Background intensity was subtracted, and the intensity values of the region of interest are reported relative to the pre-bleached images during image acquisition. Each data point represents the mean and standard error of fluorescence intensities in more than three unbleached (control) or bleached (experimental) spots.

## Live-cell imaging

Living imaging was conducted with the Olympus SpinSR10 spinning-disk confocal super-resolution microscope equipped with a Hamamatsu ORCA-Flash 4.0 camera, a UPLAPO OHR 100 × (NA 1.50) lens, and the SORA disk in place. The microscope was operated with Olympus cellSens Dimension 2.3 software. Cells were passaged onto a four-chamber glass-bottom dish (Greiner Bio-One) more than 24 h before imaging. To induce mitophagy, cells were incubated with 10 µM CCCP in the presence of 200 nM SF650-conjugated Halo ligand (GoryoChemical, A308-02). Images were processed using the image processing software package Fiji (Schindelin et al, 2012). The fluorescence intensity of indicated fluorophores was measured by Fiji, and spline-connected graphs were created using OriginPro 2022 software.

## Immunofluorescence imaging

Cells were fixed with 4% PFA in 0.1 M phosphate buffer (pH 7.3) for 15 min at room temperature (RT) and washed with PBS. Fixed cells were incubated with 10 µg/mL digitonin for 5 min at RT and washed with PBS. Then, cells were incubated with primary antibodies diluted (1:1000) in blocking buffer (3% BSA, in PBS) for 1 h at RT, washed with PBS, incubated with Alexa Fluor-conjugated secondary antibodies (1:1000) in blocking buffer for 60 min at RT, washed again, and mounted on coverslips. Colocalization was calculated by Fiji plugins BIOP JACoP, with the threshold of the ATG9A channel fixed to 500. ATG9A-positive areas out of GFP-positive areas were calculated to obtain the overlapping areas.

## Mathematical model of autophagy adaptor condensate formation

We formulated a mathematical model of autophagy adaptor condensate formation. For the case of two mitochondria approaching each other (Fig. 3A), the free energy, *F*, of the system can be written as

$$F = \int \left( f_{\text{int}} + f_{\text{ent}} \right) dV, \tag{1}$$

with the interaction energy, $f_{int}$, and the entropic energy, $f_{ent}$, expressed as

$$f_{int} = \frac{\sigma}{2}(\nabla\phi_{ad})^2 - \chi_{self}\phi_{ad}^2 - \chi_{Ub}\phi_{Ub}\phi_{ad} + \chi_{Mt}\phi_{Mt}\phi_{ad}, \quad (2)$$

$$f_{ent} = \phi_{ad}\ln\phi_{ad} + \phi_{sol}\ln\phi_{sol}, \quad (3)$$

where $\phi_{ad}$, $\phi_{Ub}$, and $\phi_{sol} = 1 - \phi_{ad} - \phi_{Ub} - \phi_{Mt}$ are the adaptor, ubiquitin, and solvent concentrations, respectively, $\chi_{self}$ and $\chi_{Ub}$ represent the strength of the adaptor self-interaction and adaptor–ubiquitin interaction, respectively, $\sigma$ represents the surface tension of the condensates, and $\chi_{Mt}$ represents the mitochondrial area exclusion effect, introduced to prevent proteins from entering the mitochondrial interior ($\phi_{Mt}$).

For the case in which an isolation membrane surrounded a mitochondrion (Fig. 3C), the interaction energy, $f_{int}$, is instead expressed as

$$f_{int} = \frac{\sigma}{2}(\nabla\phi_{ad})^2 - \chi_{self}\phi_{ad}^2 - \chi_{Ub}\phi_{Ub}\phi_{ad} - \chi_{LC3}\phi_{LC3}\phi_{ad} + \chi_{Mt}\phi_{Mt}\phi_{ad} + \chi_{IM}\phi_{IM}\phi_{ad}, \quad (4)$$

where $\phi_{LC3}$ and $\chi_{LC3}$ represent the LC3 concentrations and the strength of the adaptor–LC3 interaction, respectively, and $\chi_{IM}$ represents the mitochondrial area exclusion effect, introduced to prevent proteins from entering the lumen of the isolation membrane ($\phi_{IM}$). The solvent concentration that appears in the entropic energy (3) was also modified to $\phi_{sol} = 1 - \phi_{ad} - \phi_{Ub} - \phi_{LC3} - \phi_{Mt} - \phi_{IM}$.

For the numerical simulation, the space was discretized by a lattice with 10 nm on each side, and a concentration field was assigned to each location. The time evolution of the system was assumed to follow the Cahn–Hilliard equation,

$$\frac{\partial\phi_{ad}}{\partial t} = \frac{\partial}{\partial\vec{x}}\left[M\frac{\partial}{\partial\vec{x}}\left(\frac{\delta F}{\delta\phi_{ad}}\right)\right], \quad (5)$$

which is commonly used in analyses of phase separation dynamics (Berry et al, 2018). Here, $M = D/k_B T$ is the mobility of the protein with the diffusion constant $D = 10\ \mu m^2/sec$ (Milo and Phillips, 2015) expressed in terms of the Boltzmann constant $k_B$ and temperature $T$. We considered a 3000 nm × 3000 nm square region with periodic boundary conditions containing two mitochondria and surrounding cytosolic components as a system and assumed that adaptor proteins are uniformly distributed in the cytosol around mitochondria at a concentration of $\phi = 0.1$ in the initial state.

## Statistical analysis

Statistical analysis was performed using OriginPro 2022 software. The statistical methods used for each analysis are specified in the figure legends. No sample size calculation was performed, the experiments were not randomised, and investigators were not blinded during experiments.

## Data availability

This study includes no data deposited in external repositories.

The source data of this paper are collected in the following database record: biostudies:S-SCDT-10_1038-S44318-024-00272-5.

## Peer review information

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

## Acknowledgements

We thank Michael Lazarou (Walter and Eliza Hall Institute of Medical Research) for providing penta KO HeLa cells, Toshio Kitamura for pMXs-IP (The University of Tokyo), Shoji Yamaoka (Tokyo Medical and Dental University, Tokyo, Japan) for pMRX-IP, and Teruhito Yasui (National Institutes of Biomedical Innovation, Health and Nutrition (NIBIOHN), Osaka, Japan) for pCG-VSV-G and pCG-gag-pol. pKanCMV-mClover3-mRuby3 was a gift from Michael Lin (Addgene plasmid # 74252; http://n2t.net/addgene:74252; RRID:Addgene_74252).

This work was supported by the Exploratory Research for Advanced Technology (ERATO) research funding programme of the Japan Science and Technology Agency (JST) (JPMJER1702 to NM), a Grant-in-Aid for Specially Promoted Research (22H04919 to NM) and a Grant-in-Aid for Scientific Research (C) (23K05715 to YS) from the Japan Society for the Promotion of Science (JSPS), World Research Hub (WRH) Programme of the International Research Frontiers Initiative, Tokyo Institute of Technology (RLK), and German Research Foundation (Project 460056461 to RLK). The numerical computations have been performed with the RIKEN supercomputer system (HOKUSAI).

## Author contributions

**Zi Yang**: Conceptualisation; Formal analysis; Investigation; Visualisation; Writing—original draft. **Saori R Yoshii**: Conceptualisation; Supervision; Visualisation; Writing—original draft; Project administration. **Yuji Sakai**: Formal analysis; Funding acquisition; Investigation; Visualisation; Methodology; Writing—original draft. **Jing Zhang**: Formal analysis; Investigation; Writing—review and editing. **Haruka Chino**: Conceptualisation; Supervision; Writing—review and editing. **Roland L Knorr**: Conceptualisation; Funding acquisition; Writing—original draft. **Noboru Mizushima**: Conceptualisation; Supervision; Funding acquisition; Writing—original draft; Project administration.

Source data underlying figure panels in this paper may have individual authorship assigned. Where available, figure panel/source data authorship is listed in the following database record: biostudies:S-SCDT-10_1038-S44318-024-00272-5.

## Disclosure and competing interests statement

The authors declare no competing interests.

# Expanded View Figures

**Figure EV1.  Localisation of autophagy adaptors between mitochondria during Parkin-mediated mitophagy.**

(A) Representative images (left) and spline graphs of the intensity profiles along the indicated arrows (from position 0; right) of wild-type HeLa cells expressing one of the GFP-tagged autophagy adaptors or ubiquitin and mRuby–Omp25 at 45 min after CCCP treatment. Mitochondrial clusters (Mt–Mt) are shown. The *y*-axis in each of the graphs indicates the fluorescence intensity. (B) The relative intensity of each adaptor was calculated as in Fig. 1C. Solid horizontal bars indicate the means, and dots indicate the data from five structures. Differences were statistically analyzed by one-way ANOVA with Dunnett's post-hoc test. (C) Representative images (left) and spline graphs of the intensity profiles along the indicated arrows (from position 0; right) of HeLa cells lacking all five autophagy adaptors (penta KO cells) expressing one of the GFP-tagged autophagy adaptors or ubiquitin and mRuby–Omp25 at 45 min after CCCP treatment. Mitochondrial clusters (Mt–Mt) are shown. The *y*-axis in each of the graphs indicates the fluorescence intensity. (D) The relative intensity of each adaptor was calculated as in Fig. 1C. Solid horizontal bars indicate the means, and dots indicate the data from five structures. Differences were statistically analyzed by one-way ANOVA with Dunnett's post-hoc test. Source data are available online for this figure.

▶

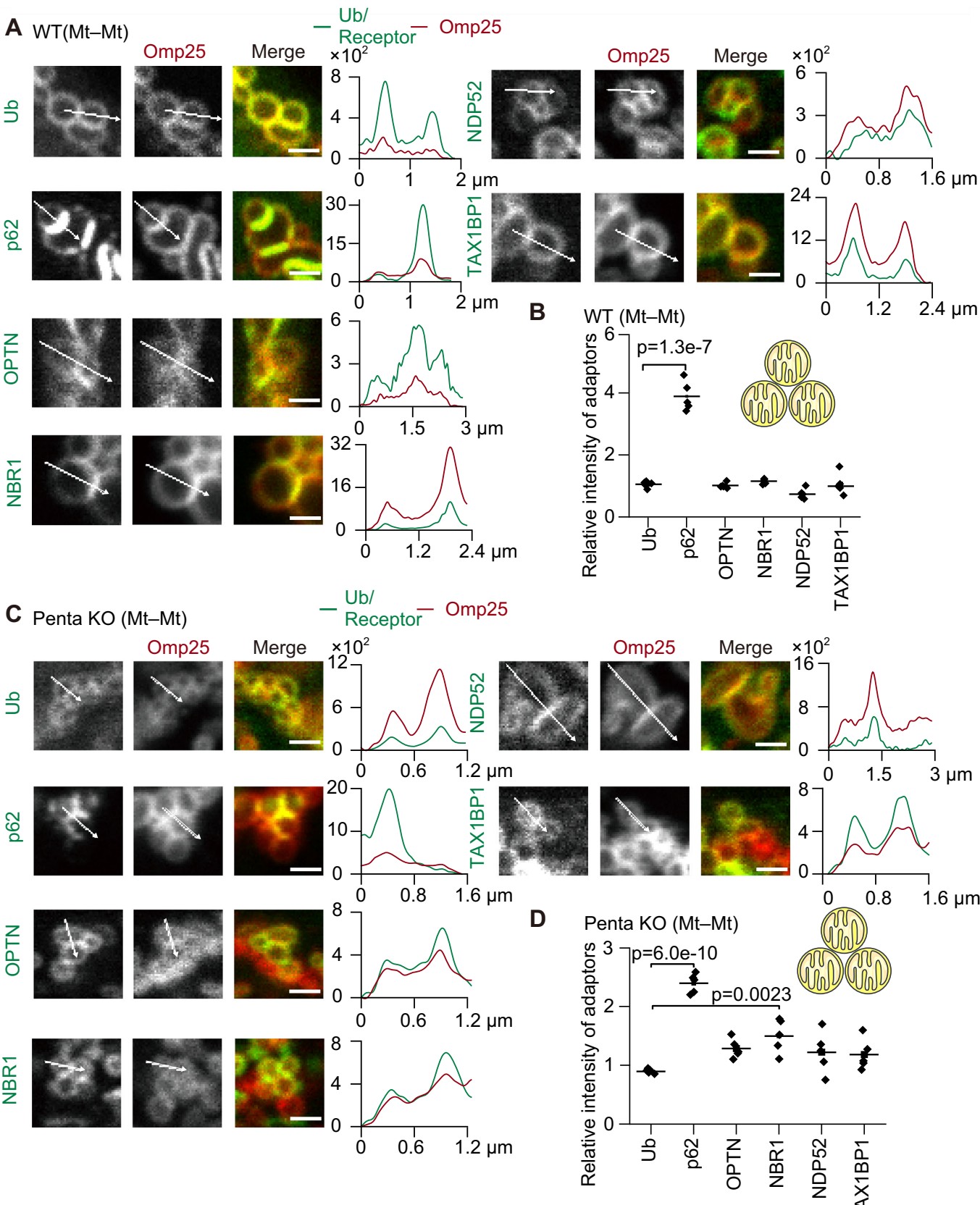

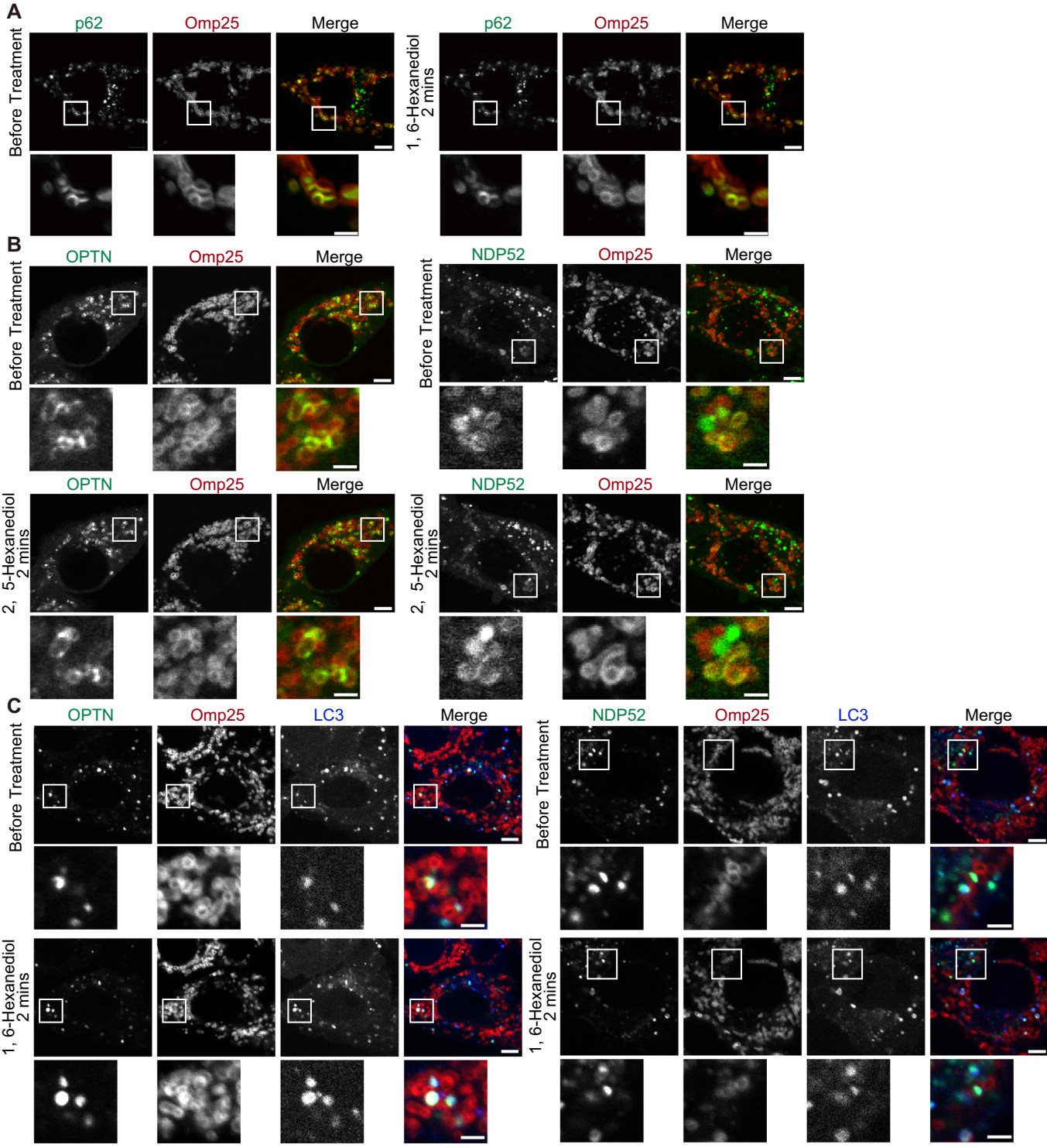

**Figure EV2.    p62 on mitochondrial clusters and OPTN and NDP52 at the mitochondria-isolation membrane contact sites are resistant to 1,6-hexanediol treatment.**

(**A**) Penta KO cells expressing GFP–p62 at 45 min after CCCP treatment. Images of cells before (left) and 2 min after (right) the addition of 10% 1,6-hexanediol are displayed. Scale bars, 5 and 2 μm (magnified images). (**B**) Penta KO cells expressing GFP–OPTN (left) or GFP–NDP52 (right) at 45 min after CCCP treatment. Images of cells before (upper panels) and 2 min after (lower panels) the addition of 10% 2,5-hexanediol are displayed. Scale bars, 5 and 2 μm (magnified images). (**C**) Penta KO cells expressing GFP–OPTN (left) or GFP–NDP52 (right) at 45 min after CCCP treatment. Mitochondria associated with isolation membranes (LC3) were analyzed. Images of cells before (upper panels) and 2 min after (lower panels) the addition of 10% 1,6-hexanediol are displayed. Scale bars, 5 and 2 μm (magnified images). Source data are available online for this figure.

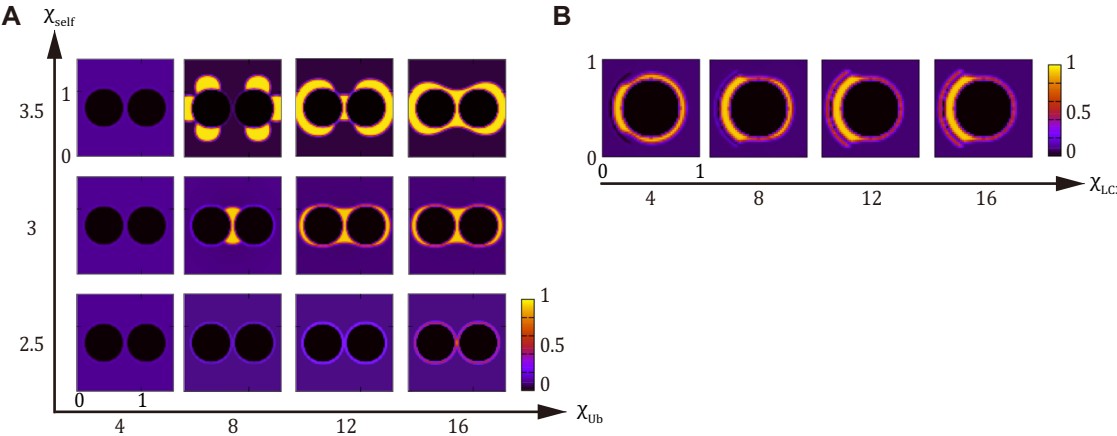

**Figure EV3. Droplet distribution depends on the binding strength.**

(A) Dependence of droplet distribution on two mitochondria on Ub-binding strength ($\chi_{Ub}$) and self-binding strength ($\chi_{self}$). Mitochondrial area exclusion ($\chi_{Mt}$) and surface tension ($\sigma$) were set to $\chi_{Mt} = 5k_BT$ and $\sigma = k_BT$, respectively. (B) Dependence of droplet distribution on isolation membrane and mitochondria on LC3-binding strength ($\chi_{LC3}$). The other parameters were set to $\chi_{Mt} = 5k_BT$, $\sigma = k_BT$, $\chi_{self} = 3k_BT$ and $\chi_{Ub} = 12k_BT$. The bending angle of the isolation membrane was set to $\alpha = \pi/3$.

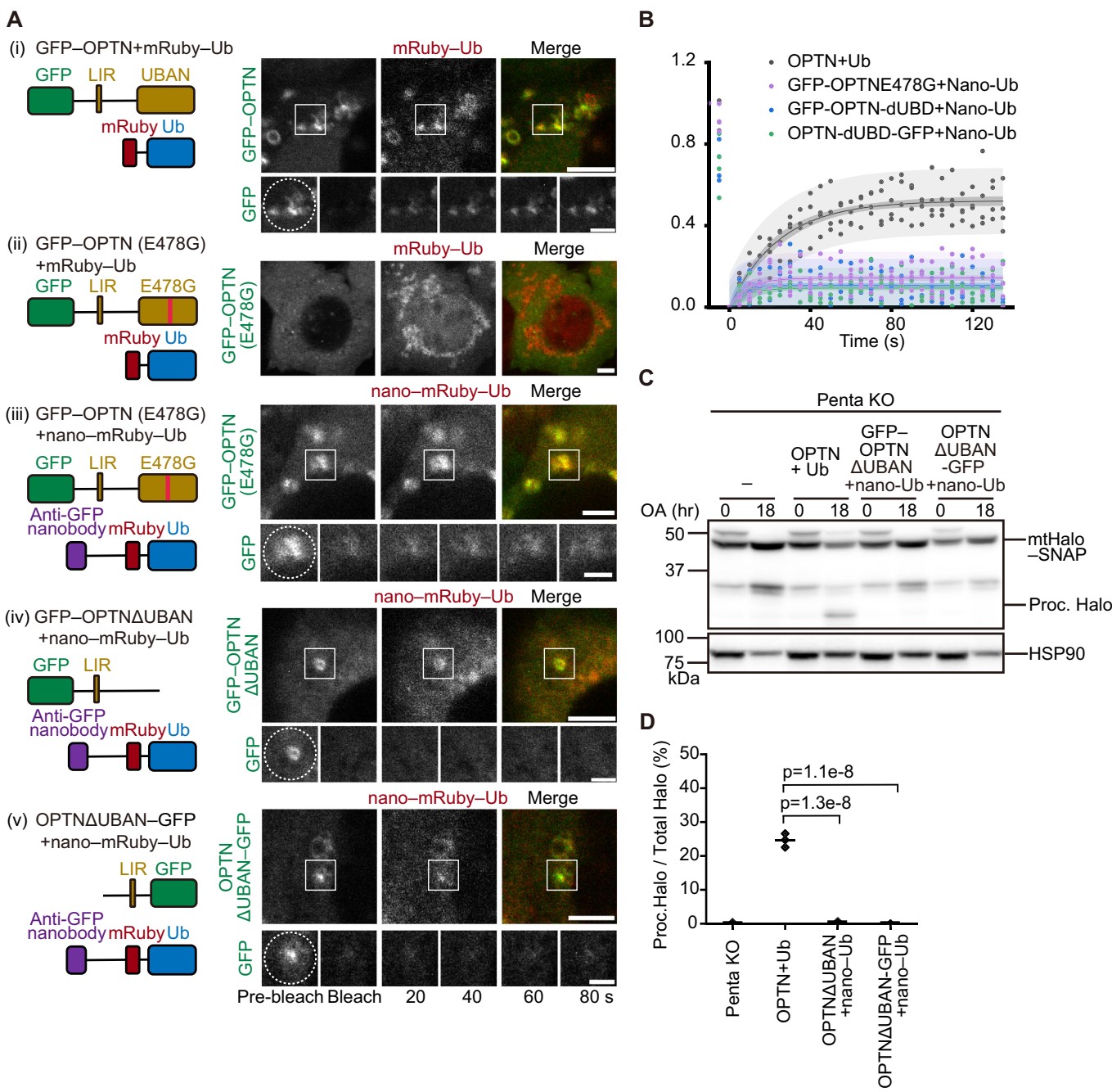

**Figure EV4. Loss of mobility and mitophagy activity with ubiquitin binding mutants of OPTN.**

(A) Penta KO cells expressing both GFP–OPTN and mRuby–Ub (i), GPF–OPTN (E478G) and mRuby–Ub (ii), both GPF–OPTN (E478G) and anti-GFP nanobody–mRuby–Ub (iii), both GFP–OPTNΔUBAN and anti-GFP nanobody–mRuby–Ub (iv), and both OPTNΔUBAN–GFP and anti-GFP nanobody–mRuby–Ub (v) at 45 min after CCCP treatment. Time-lapse images of GFP FRAP are shown. Photobleached areas are circled by dotted lines. Scale bars indicate 4 and 2 µm (magnified images). (B) Quantification of GFP FRAP on separate mitochondria (Mt) in penta KO cells expressing both GFP–OPTN and mRuby–Ub (i), both GPF–OPTN (E478G) and anti-GFP nanobody–mRuby–Ub (iii), both GFP–OPTNΔUBAN and anti-GFP nanobody–mRuby–Ub (iv), and both OPTNΔUBAN–GFP and anti-GFP nanobody–mRuby–Ub (v) at 45 min after CCCP treatment. Data were collected from four structures and were fitted to the equation y = a*(1 − exp(−b*x)). The dark shading represents the 95% confidence intervals, and the light shading represents the 95% prediction intervals. (C, D) Representative data (C) and quantification (D) of HaloTag (Halo) processing assay using cells expressing the indicated OPTN and Ub constructs. Cells expressing the mtHalo–SNAP mitophagy reporter were treated without (0 h) and with oligomycin and antimycin for 18 h. The amount of processed Halo (proc. Halo) indicates the relative amount of mitochondria degraded in lysosomes. Solid horizontal bars indicate the means, and dots indicate the data from three independent cultures. Differences were statistically analyzed by one-way analysis of variance with Dunnett's post-hoc test. Source data are available online for this figure.

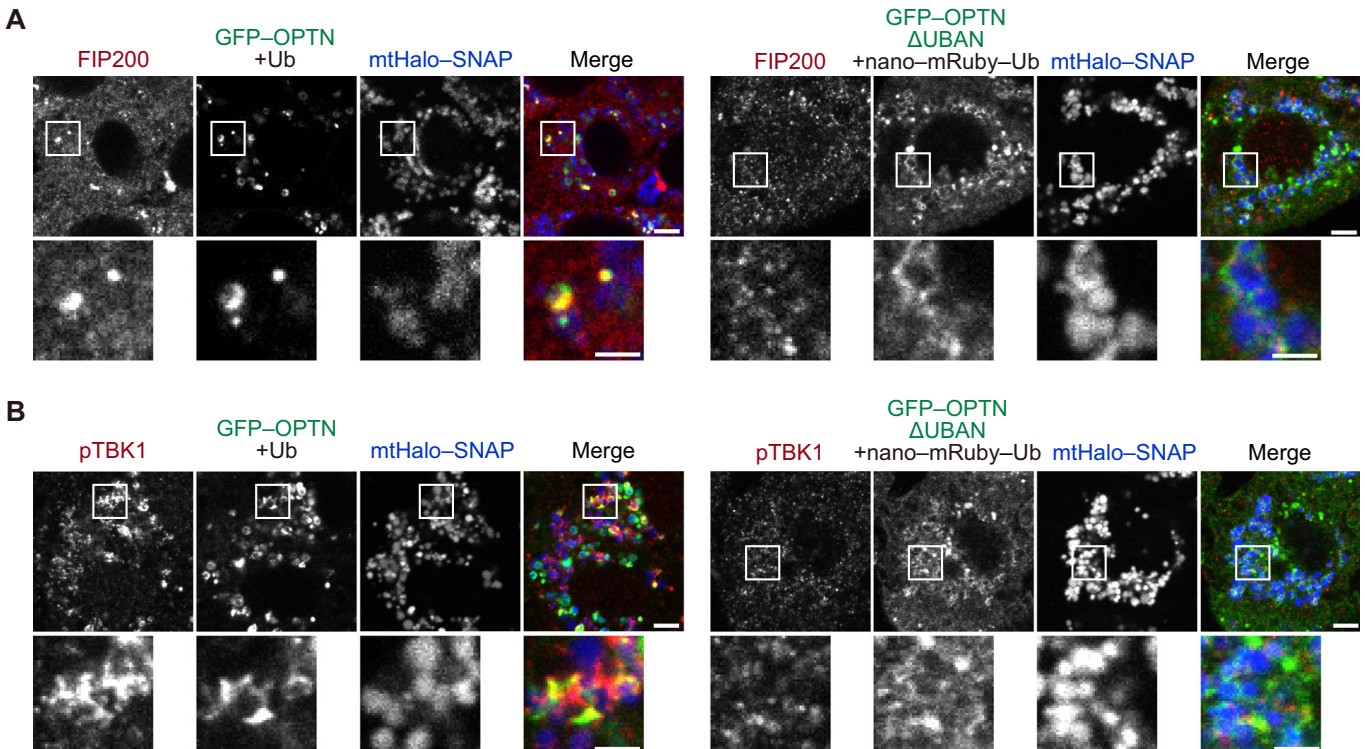

**Figure EV5.  Recruitment of TBK1 and FIP200 during mitophagy.**

(A, B) Localisation of FIP200 and phosphorylated TBK1 under mitophagy-inducing conditions (CCCP, 60 min). Endogenous FIP200 (A) and phosphorylated TBK1 (B) were immunostained in penta KO cells expressing both GFP–OPTN and mRuby–Ub or both GFP–OPTNΔUBAN and anti-GFP nanobody–mRuby–Ub together with mitochondrially targeted Halo-SNAP (mtHalo–SNAP). Scale bars indicate 4 and 2 μm (magnified images). Source data are available online for this figure.

