## [Peer Review File · The EMBO Journal]

Autophagy adaptors mediate Parkin-dependent mitophagy by forming sheet-like liquid condensates

Zi Yang, Saori Yoshii, Yuji Sakai, Jing Zhang, Haruka Chino, Roland Knorr, and Noboru Mizushima

Corresponding author(s): Noboru Mizushima (nmizu@m.u-tokyo.ac.jp)

Review Timeline:

Submission Date:	8th Sep 23
Editorial Decision:	2nd Nov 23
Revision Received:	13th Aug 24
Editorial Decision:	4th Sep 24
Revision Received:	20th Sep 24
Accepted:	29th Sep 24

Editor: William Teale

Transaction Report:

Dear Noboru,

Thank you again for the submission of your manuscript entitled "Autophagy adaptors mediate Parkin-dependent mitophagy by forming sheet-like liquid condensates" and for your patience during the review process. We have now received reports from three referees, which I copy below.

As you can see from their comments, while the referees make some suggestions on how you might consider strengthening the manuscript, all are broadly supportive of your experimental approach.

Therefore, based on the overall interest expressed in the reports, I would like to invite you to address the comments of all referees in a revised version of the manuscript. I should add that it is The EMBO Journal policy to allow only a single major round of revision and that it is therefore important to resolve the main concerns at this stage. I believe the concerns of the referees are reasonable and addressable, but please contact me if you have any questions, need further input on the referee comments or if you anticipate any problems in addressing any of their points. Please, follow the instructions below when preparing your manuscript for resubmission.

I would also like to point out that as a matter of policy, competing manuscripts published during this period will not be taken into consideration in our assessment of the novelty presented by your study ("scooping" protection). We have extended this 'scooping protection policy' beyond the usual 3 month revision timeline to cover the period required for a full revision to address the essential experimental issues. Please contact me if you see a paper with related content published elsewhere to discuss the appropriate course of action.

Again, please contact me at any time during revision if you need any help or have further questions.

Thank you very much again for the opportunity to consider your work for publication. I look forward to your revision.

Best regards,

William

William Teale, Ph.D.
Editor
The EMBO Journal

When submitting your revised manuscript, please carefully review the instructions below and include the following items:

- 1) a .docx formatted version of the manuscript text (including legends for main figures, EV figures and tables). Please make sure that the changes are highlighted to be clearly visible.
- 2) individual production quality figure files as .eps, .tif, .jpg (one file per figure).
- 3) a .docx formatted letter INCLUDING the reviewers' reports and your detailed point-by-point response to their comments. As part of the EMBO Press transparent editorial process, the point-by-point response is part of the Review Process File (RPF), which will be published alongside your paper.
- 4) a complete author checklist, which you can download from our author guidelines ([https://wol-prod-cdn.literatumonline.com/pb-assets/embo-site/Author Checklist%20-%20EMBO%20J-1561436015657.xlsx](https://wol-prod-cdn.literatumonline.com/pb-assets/embo-site/Author%20Checklist%20-%20EMBO%20J-1561436015657.xlsx)). Please insert information in the checklist that is also reflected in the manuscript. The completed author checklist will also be part of the RPF.
- 5) Please note that all corresponding authors are required to supply an ORCID ID for their name upon submission of a revised manuscript.
- 6) We require a 'Data Availability' section after the Materials and Methods. Before submitting your revision, primary datasets produced in this study need to be deposited in an appropriate public database, and the accession numbers and database listed under 'Data Availability'. Please remember to provide a reviewer password if the datasets are not yet public (see <https://www.embopress.org/page/journal/14602075/authorguide#datadeposition>). If no data deposition in external databases is

needed for this paper, please then state in this section: This study includes no data deposited in external repositories. Note that the Data Availability Section is restricted to new primary data that are part of this study.

Note - All links should resolve to a page where the data can be accessed.

8) For data quantification: please specify the name of the statistical test used to generate error bars and P values, the number (n) of independent experiments (specify technical or biological replicates) underlying each data point and the test used to calculate p-values in each figure legend. The figure legends should contain a basic description of n, P and the test applied. Graphs must include a description of the bars and the error bars (s.d., s.e.m.).

9) We would also encourage you to include the source data for figure panels that show essential data. Numerical data can be provided as individual .xls or .csv files (including a tab describing the data). For 'blots' or microscopy, uncropped images should be submitted (using a zip archive or a single pdf per main figure if multiple images need to be supplied for one panel). Additional information on source data and instruction on how to label the files are available at .

10) We replaced Supplementary Information with Expanded View (EV) Figures and Tables that are collapsible/expandable online (see examples in <https://www.embopress.org/doi/10.15252/embj.201695874>). A maximum of 5 EV Figures can be typeset. EV Figures should be cited as 'Figure EV1, Figure EV2" etc. in the text and their respective legends should be included in the main text after the legends of regular figures.

12) Our journal encourages inclusion of *data citations in the reference list* to directly cite datasets that were re-used and obtained from public databases. Data citations in the article text are distinct from normal bibliographical citations and should directly link to the database records from which the data can be accessed. In the main text, data citations are formatted as follows: "Data ref: Smith et al, 2001" or "Data ref: NCBI Sequence Read Archive PRJNA342805, 2017". In the Reference list, data citations must be labeled with "[DATASET]". A data reference must provide the database name, accession number/identifiers and a resolvable link to the landing page from which the data can be accessed at the end of the reference. Further instructions are available at .

Further instructions for preparing your revised manuscript:

We realize that it is difficult to revise to a specific deadline. In the interest of protecting the conceptual advance provided by the work, we recommend a revision within 3 months (31st Jan 2024). Please discuss the revision progress ahead of this time with the editor if you require more time to complete the revisions. Use the link below to submit your revision:

Referee #1:

OPTN and NDP52 a key mediators of PINK1 and Parkin mitophagy and also play fundamental roles as selective autophagy cargo adaptors. Zang et al discover that OPTN and NDP52 form condensates on the surface of mitochondria that are concentrated at the interface of phagophores and mitochondria during mitophagy. The formation of condensates by OPTN was shown to be important for mitophagy activity, including stable recruitment of ATG9A vesicles to mitochondria. This is a very interesting discovery that provides an important mechanistic advance in understanding ubiquitin dependent selective autophagy. Overall, this is an excellent study, the data are very clear and convincing, and the findings are of broad interest and therefore highly suitable for the readership of EMBO J. Below are some suggestions for the authors to consider to help further strengthen their conclusions and also provide further mechanistic insight into the role of OPTN condensates in mitophagy.

Major:

1. Figure 5D and 6A: The defect in mitophagy and recruitment of ATG9A are clear. However, it would be beneficial to analyse and quantitate additional autophagy markers to determine whether ATG9A vesicle recruitment is the only/primary defect. This includes the ULK1 complex (e.g. ATG13), and/or the PI3K complex, as well as the recruitment and activation of OPTN's partner, TBK1 (using anti TBK1 and/or anti-phospho TBK1 antibodies).
2. Given that ATG8 proteins may also contribute to concentrating adaptors at the phagophore side of mitophagy events, it would be interesting to conduct the nanobody experiment with an ATG8 member, e.g. LC3B, to address whether it has the same effect as ubiquitin. This would help strengthen the authors conclusions and could also help to determine whether the liquidity can also contribute after phagophore initiation. It is possible that it only functions to recruit ATG9A during early stages, or that it can also function downstream of ATG9A recruitment following ATG8 lipidation.

Minor:

1. Abstract, lines 27 and 59: Suggested rewording away from using the term bridging: autophagy adaptors function by binding to ubiquitinated mitochondria and autophagy initiation machineries, rather than bridging via ATG8 family members. ATG8 proteins are not critical for bridging during PINK1-Parkin mitophagy, but they do amplify mitophagy signalling by contributing to the enrichment of OPTN and NDP52 at sites of autophagosome formation (Padman et al (2019) Nat Commun), which may explain the contribution of the phagophore membrane side to condensate formation (see comment 2 above), while ubiquitin contributes on the mitochondrial membrane side.

2. Lines 24 and 50: Loss of membrane potential is not a pre-requisite to trigger PINK1 and Parkin mitophagy since damaged and polarized mitochondria can undergo mitophagy as well (e.g. Jin & Youle (2013) *Autophagy*, Benjamin Michaelis (2022) *Nat Commun*), it therefore might be better to use the term damaged mitochondria

3. Line 60: In addition to ATG9A binding, OPTN also recruits the PI3K complex via its interaction with TBK1 (Nguyen et al (2023) *Mol Cell*) which is essential for mitophagy initiation.

4. Line 124: A more suitable study for this statement is: Narendra et al (2010) *Autophagy*

5. Figure 5B and Line 238: the authors state that the binding of OPTN to ub is significantly enhanced in the presence of nano-ub in Fig 3B, but the experiment was only conducted for OPTN lacking its UBAN. The statement either needs rewording or experimental evidence showing full length OPTN with and without nano-Ub to compare would be necessary.

6. The requirement for condensates in mitophagy was demonstrated for OPTN, and while it may also be the case for NDP52, this was not directly shown. Therefore, it would be worthwhile to briefly discuss in the discussion along the lines that future work will be required to establish whether the condensates formed by NDP52 are also necessary for its mitophagy activity.

Referee #2:

The manuscript discusses the role of liquid-liquid phase separation (LLPS) in autophagy, specifically focusing on the autophagy receptor protein Optineurin (OPTN) and its involvement in selective mitophagy. The authors provide evidence that autophagy adaptors, including OPTN, undergo LLPS processes that facilitate mitophagy by concentrating in regions of the mitochondrial membrane where LC3B is present. The liquidity of OPTN condensates is found to be crucial for mitophagy activity.

Previous publications have discussed LLPS in the context of autophagy. For instance, LLPS occurs when p62/SQSTM1 binds to ubiquitinated substrate proteins, leading to phagophore formation and autophagosome formation (PMID: 29507397). Parkin, an E3 ubiquitin ligase, can be activated by PINK1 and interact with E2 ubiquitin-coupled enzymes through LLPS to mediate ubiquitination of mitochondrial outer membrane substrate proteins. Although OPTN has been shown to mediate phagophore and autophagosome formation through interacting with ATG9a, using fluoppi assay, (PMID: 32556086). The actual phase separation of OPTN at native state was understated. This paper represents the first example showing that OPTN mediate mitophagy through phase separation. Overall, this study advanced our understanding of the mechanisms underlying mitophagy.

There are a few concerns regarding the manuscript.

1. The authors have introduced a mathematical model that describes the dynamic interplay between autophagic receptors and membranes during Parkin-mediated mitophagy. However, there are several issues with the model. The parameter set used in the model is based on ideal conditions, and the validation of the model is incomplete.
2. The causal relationship between the liquid properties of the condensates and autophagy function is not clearly demonstrated. For example, in Figure 5A, the authors used the mathematical model to predict the distribution of OPTN during mitochondrial-mitochondrial (mt-mt) interaction in cells expressing GFP-OPTN Δ UBAN with nanobody-mRuby-Ub. Are there any mutants with lower liquidity of OPTN that suppress its accumulation in the cleft between mitochondria?
3. In Figure 2E:
 - (a) The treatment with 1,6-hexanediol disperses OPTN from mitochondria without isolation membranes, how about the interaction between OPTN and LC3? Would it hinder the dispersion?
 - (b) The 2,5-hexanediol is encouraged to be included as a control, will it also disperse OPTN from mitochondria?
 - (c) Could NDP52 condensates be dispersed after 1,6-hexanediol treatment?
4. In Figure 3A, the authors assume that the adaptor proteins (OPTN) diffuse freely in solution to fill the cleft between mitochondria, which forms the basis of the mathematical model. However, in Figure EV1A, p62 also accumulate between clustered mitochondria, despite the FRAP results in Figure 2A showing slow recovery of p62 condensates. How is the mobility of these adaptors correlate with their actual functions? Further details should be included to further support the mathematical model.
5. In Figure 5A, will the mobility of GFP-OPTN be affected by nanobody-mRuby-Ub, and further, mitophagy activity?

Referee #3:

In the manuscript by Yang et al., the authors probe the role of autophagy receptors in forming liquid condensates during Parkin-

dependent mitophagy. Using a combination of cell biology and mathematical modelling, the authors find that OPTN and NDP52 associate with mitochondria in a dynamic fashion following mitophagy activation and redistribute their mitochondrial localization to sites of isolation membrane initiation. This is not seen with the other autophagy receptors, including p62. Further characterizing OPTN, the authors go on to show that strengthening its interaction with ubiquitin (via substituting the OPTN UBAN domain with an anti-GFP nanobody system) results in mitochondrial recruitment following stimulation but a loss in fluid-like dynamics and a failure to induce mitophagy. Finally, the authors demonstrate that the mitochondrially associated OPTN UBAN mutant does not recruit ATG9 to mitochondria to initiate mitophagy.

Overall this is a very interesting manuscript and potentially brings key knowledge about the role and mechanism that OPTN plays in initiating mitophagy. The authors have demonstrated the LLPS nature of OPTN on mitochondria, but I feel the key experimental data showing that this is critical for mitophagy need strengthening.

Main Points:

1) In Figure 2E, as a control, does 1,6-Hexanediol affect p62 staining on mitochondria in a similar manner to OPTN? Given the results in panel A, is it more stable than OPTN?

2) The data in Figure 5 are key in defining OPTN LLPS as a key mechanism for mitophagy. However, is it possible that the orientation/conformation of OPTN (upon mitochondrial recruitment with the nanobody) that is restricted and not LLPS per se? Does replacement of the UBAN domain with GFP at the C-terminus also block mitophagy?

3) Related to the above, loss of the C-terminal domain of OPTN, could be considered an extreme modification to the protein. Is this construct phosphorylated by TBK1 upon mitophagy initiation? Are similar observations also seen with just a point mutant of OPTN that impairs ubiquitin binding (such as the ALS-related E478G mutation)?

4) The data suggests that affinity of receptor for ubiquitin binding is important for mitochondrial redistribution to sites of isolation membrane initiation. Given that p62 is recruited to ubiquitinated mitochondria but does not redistribute, is anything known about the ubiquitin-binding affinities of p62's UBA domain compared to OPTN's UBAN domain: does substitution of the UBAN domain on OPTN with p62's UBA domain result in a block in OPTN redistribution and mitophagy?

Responses to the comments of Reviewer #1

OPTN and NDP52 are key mediators of PINK1 and Parkin mitophagy and also play fundamental roles as selective autophagy cargo adaptors. Zang et al discover that OPTN and NDP52 form condensates on the surface of mitochondria that are concentrated at the interface of phagophores and mitochondria during mitophagy. The formation of condensates by OPTN was shown to be important for mitophagy activity, including stable recruitment of ATG9A vesicles to mitochondria. This is a very interesting discovery that provides an important mechanistic advance in understanding ubiquitin dependent selective autophagy. Overall, this is an excellent study, the data are very clear and convincing, and the findings are of broad interest and therefore highly suitable for the readership of EMBO J. Below are some suggestions for the authors to consider to help further strengthen their conclusions and also provide further mechanistic insight into the role of OPTN condensates in mitophagy.

We appreciate Reviewer #1 for the positive evaluation and valuable suggestions. In response, we have added several key datasets and revised the manuscript accordingly.

Major:

1. Figure 5D and 6A: The defect in mitophagy and recruitment of ATG9A are clear. However, it would be beneficial to analyse and quantitate additional autophagy markers to determine whether ATG9A vesicle recruitment is the only/primary defect. This includes the ULK1 complex (e.g. ATG13), and/or the PI3K complex, as well as the recruitment and activation of OPTN's partner, TBK1 (using anti TBK1 and/or anti-phospho TBK1 antibodies).

We thank the reviewer for the valuable suggestion. We stained phospho-TBK1 and FIP200 (as a component of the ULK1 complex) to analyze their colocalization with mitochondria and OPTN under mitophagy-inducing conditions. p-TBK1 showed clear accumulation around mitochondria in the presence of wild-type GFP–OPTN but only very slight accumulation in the presence of GFP–OPTN Δ UBAN and nanobody–ubiquitin (Fig. EV5B). These data suggest that the condensate formation contributes to the positive feedback

loop between the OPTN accumulation and TBK1 activation, which was recently proposed by the Lazarou and Matsuda groups (Nguyen et al., Mol Cell 2023, Yamano et al., EMBO J 2024). These data are now described on page 13, lines 324-331.

A similar decrease in signals was observed for FIP200 (Fig. EV5A). Since the ULK1 complex is dispensable for OPTN–TBK1-dependent mitophagy (Nguyen et al., Mol Cell 2023), the failure to recruit the ULK1 complex is NOT likely to be the cause of the mitophagy defect. Instead, we suspect that the lack of FIP200 puncta is due to the failure of isolation membrane formation, downstream of the ATG9 complex recruitment. These data are now described on page 12, lines 293-298

2. Given that ATG8 proteins may also contribute to concentrating adaptors at the phagohore side of mitophagy events, it would be interesting to conduct the nanobody experiment with an ATG8 member, e.g. LC3B, to address whether it has the same effect as ubiquitin. This would help strengthen the authors conclusions and could also help to determine whether the liquidity can also contribute after phagophore initiation. It is possible that it only functions to recruit ATG9A during early stages, or that it can also function downstream of ATG9A recruitment following ATG8 lipidation.

We appreciate the reviewer's insightful suggestion; we agree that this is a very interesting point to address. We tested mitophagy activity and protein localization in cells expressing nanobody–LC3 and GFP–OPTN Δ LIR. Interestingly, CoxIV degradation appeared to be reduced, implying that the liquidity of the condensates or reversible interactions between ubiquitinated mitochondria, OPTN, and isolation membrane support isolation membrane elongation (Response Figure 1A-C). However, nanobody–LC3 was uniformly recruited to all GFP–OPTN Δ LIR-positive structures (Response Figure 1A), likely because of the strong interaction between soluble unlipidated nanobody–LC3-I and GFP–OPTN Δ LIR. For this reason, we were unable to identify isolation membrane-positive mitochondria for FRAP analysis to test the liquidity of OPTN between mitochondria and isolation membranes. Although defective isolation membrane formation is suggested, we cannot conclude that it is due to a decrease in liquidity and would like to refrain from showing data in the manuscript to avoid potential misinterpretation.

Response Figure 1. Expressing Halo–nanobody–LC3 with OPTNΔLIR cannot rescue mitophagy.

(A) Localization of OPTN or OPTNΔLIR under mitophagy-inducing conditions (CCCP+, 60 min). Scale bars indicate 4 μm and 2 μm (magnified images).

(B and C) Representative data (B) and quantification (C) of mitophagy activity using cells expressing the indicated OPTN and LC3 constructs. Cells were treated without (0 h) and with 1 μM oligomycin and 2 μM antimycin for 18 h. The amount of remaining COX IV indicates the amount of mitochondria not degraded by the lysosomes. Solid horizontal bars indicate the means, and dots indicate the data from four independent experiments. Differences were statistically analyzed by one-way analysis of variance with Dunnett’s post-hoc test.

Minor:

1. Abstract, lines 27 and 59: Suggested rewording away from using the term bridging: autophagy adaptors function by binding to ubiquitinated mitochondria and autophagy initiation machineries, rather than bridging via ATG8 family members. ATG8 proteins are not critical for bridging during PINK1-Parkin mitophagy, but they do amplify mitophagy signaling by contributing to the enrichment of OPTN and NDP52 at sites of autophagosome formation (Padman et al (2019) Nat Commun), which may explain the contribution of the phagophore membrane side to condensate formation (see comment 2 above), while ubiquitin contributes on the mitochondrial membrane side.

We thank the reviewer for the valuable suggestion. In response, we revised the abstract to state, “OPTN and NDP52 facilitate mitophagy by recruiting autophagy initiation machinery and assisting the engulfment of damaged mitochondria” (page 2, lines 30-31). Furthermore, we have modified the introduction to read “supporting efficient engulfment by autophagosomes” (page 3, lines 64-65).

2. Lines 24 and 50: Loss of membrane potential is not a pre-requisite to trigger PINK1 and Parkin mitophagy since damaged and polarized mitochondria can undergo mitophagy as well (e.g. Jin & Youle (2013) *Autophagy*, Benjamin Michaelis (2022) *Nat Commun*), it therefore might be better to use the term damaged mitochondria

We thank the reviewer for the suggestion. We have now corrected the sentences accordingly (page 2, line 29; page 3, lines 55-56).

3. Line 60: In addition to ATG9A binding, OPTN also recruits the PI3K complex via its interaction with TBK1 (Nguyen et al (2023) *Mol Cell*) which is essential for mitophagy initiation.

We thank the reviewer for the suggestion. We have now cited the suggested reference to state, “OPTN recruits ATG9 vesicles and TBK1, which subsequently recruits the PI3K complex” (page 3, lines 67-69).

4. Line 124: A more suitable study for this statement is: Narendra et al (2010) *Autophagy*

As suggested by the reviewer, we have now cited the reference and mentioned: “which appears to be consistent with the role of p62 in promoting mitochondrial clustering” (page 6, lines 132-133) because the paper primarily focuses on mitochondrial clustering rather than the specific accumulation of p62 between mitochondria. We have also added another relevant paper (Okatsu et al. 2010).

5. Figure 5B and Line 238: the authors state that the binding of OPTN to ub is significantly enhanced in the presence of nano-ub in Fig 3B, but the experiment

was only conducted for OPTN lacking its UBAN. The statement either needs rewording or experimental evidence showing full length OPTN with and without nano-Ub to compare would be necessary.

We thank the reviewer for raising the critical point. We have now added data of cells expressing GFP–OPTN (wild-type) along with nanobody–ubiquitin. Indeed, nanobody–Ub expression greatly enhanced its interaction also with wild-type GFP–OPTN (Fig. 5B). Additionally, it caused partial defects in FRAP recovery and mitophagy, demonstrating an intermediate phenotype between cells expressing wild-type OPTN and those expressing nanobody–Ub and GFP–OPTN Δ UBAN (Figure 5C, D and E). We speculate that this intermediate phenotype is due to the presence of abundant endogenous ubiquitin, to which GFP–OPTN can also bind, and thereby, the condensate formation was not completely abolished. We have now added the data and described them on page 11, lines 257-259 and 268-269.

6. The requirement for condensates in mitophagy was demonstrated for OPTN, and while it may also be the case for NDP52, this was not directly shown. Therefore, it would be worthwhile to briefly discuss in the discussion along the lines that future work will be required to establish whether the condensates formed by NDP52 are also necessary for its mitophagy activity.

We appreciate the valuable suggestion by the reviewer. We have incorporated the raised point in the Discussion, stating, “Additionally, our experimental data suggest that NDP52 also exhibits liquid-like properties. Further studies are needed to ascertain whether condensate formation is necessary for mitophagy induction by NDP52” (page 14, lines 350-352).

Responses to the comments of Reviewer #2

The manuscript discusses the role of liquid-liquid phase separation (LLPS) in autophagy, specifically focusing on the autophagy receptor protein Optineurin (OPTN) and its involvement in selective mitophagy. The authors provide evidence that autophagy adaptors, including OPTN, undergo LLPS processes that facilitate mitophagy by concentrating in regions of the mitochondrial membrane where LC3B is present. The liquidity of OPTN condensates is found to be crucial for mitophagy activity.

Previous publications have discussed LLPS in the context of autophagy. For instance, LLPS occurs when p62/SQSTM1 binds to ubiquitinated substrate proteins, leading to phagophore formation and autophagosome formation (PMID: 29507397). Parkin, an E3 ubiquitin ligase, can be activated by PINK1 and interact with E2 ubiquitin-coupled enzymes through LLPS to mediate ubiquitination of mitochondrial outer membrane substrate proteins. Although OPTN has been shown to mediate phagophore and autophagosome formation through interacting with ATG9a, using fluoppi assay, (PMID: 32556086). The actual phase separation of OPTN at native state was understated. This paper represents the first example showing that OPTN mediate mitophagy through phase separation. Overall, this study advanced our understanding of the mechanisms underlying mitophagy.

We would like to thank Reviewer #2 for the positive evaluation and insightful questions. We have now revised the manuscript accordingly.

There are a few concerns regarding the manuscript.

1. The authors have introduced a mathematical model that describes the dynamic interplay between autophagic receptors and membranes during Parkin-mediated mitophagy. However, there are several issues with the model. The parameter set used in the model is based on ideal conditions, and the validation of the model is incomplete.

We apologize for the lack of validation of the model. We have included new data from the mathematical model with different Ub-, self-, and LC3-binding

strengths. The accumulation of the adaptor protein on mitochondria depended on both ubiquitin-binding and self-binding strengths (Figure EV3A). Excessive accumulation and morphological changes of the droplet were observed when self-binding was too strong ($\chi_{self} > 3k_B T$), resulting in a lack of sheet-like condensates observed in cells. These data underscore the need to balance the ub- and self-binding strengths of the autophagy adaptors. Therefore, we used the model parameters $\chi_{self} = 3k_B T$ and $\chi_{Ub} = 12k_B T$ for further analyses.

Similarly, we tested different LC3-binding strengths at $\chi_{self} = 3k_B T$ and $\chi_{Ub} = 12k_B T$. A clear accumulation of the autophagy adaptor in the region in contact with the isolation membrane, along with depletion from the non-contacting region, was observed at $\chi_{LC3} = 12k_B T$ or higher (Figure EV3B), mirroring the images observed in Figure 4E. These model data suggest that the parameters we used in the manuscript recapitulate the observed behavior of OPTN and support the plausibility of the parameters used. We have now added the data and described them on page 8-9, lines 195-198 and lines 214-217.

2. The causal relationship between the liquid properties of the condensates and autophagy function is not clearly demonstrated. For example, in Figure 5A, the authors used the mathematical model to predict the distribution of OPTN during mitochondrial-mitochondrial (mt-mt) interaction in cells expressing GFP-OPTN Δ UBAN with nanobody-mRuby-Ub. Are there any mutants with lower liquidity of OPTN that suppress its accumulation in the cleft between mitochondria?

As suggested by the reviewer, we tested the behavior of GFP-OPTN Δ UBAN during mt-mt interaction in the presence of nanobody-mRuby-Ub. We did not observe the accumulation of OPTN Δ UBAN in the cleft between two mitochondria (Response Figure 2A and B). This could be a phenotype of decreased liquidity, as anticipated by the reviewer. However, the amount of GFP-OPTN Δ UBAN on mitochondria was smaller compared to the amount accumulated with GFP-OPTN(WT) as described on page 11, lines 276-277. Thus, while the lack of liquidity could be the cause, it is also possible that we failed to capture the event due to its reduced concentration/amount. Therefore, we believe it is prudent to refrain from explicitly presenting the data in the manuscript.

Response Figure 2. Redistribution of GFP–OPTNΔUBAN is not observed upon mitochondrial contact.

Time-lapse images (A) and spline graphs (B) of two separate mitochondria with comparable sizes approaching and contacting each other in cells expressing GFP–OPTNΔUBAN and nanobody–mRuby–Ub. The line graphs represent the intensity profiles along the indicated lines in the respective images shown (B). Scale bars indicate 1 μ m.

3. In Figure 2E:

- (a) The treatment with 1,6-hexanediol disperses OPTN from mitochondria without isolation membranes, how about the interaction between OPTN and LC3? Would it hinder the dispersion?
- (b) The 2,5-hexanediol is encouraged to be included as a control, will it also disperse OPTN from mitochondria?
- (c) Could NDP52 condensates be dispersed after 1,6-hexanediol treatment?

We appreciate the reviewer’s feedback and address the points as follows:

- (a) As shown in Figure EV2C, 1,6-hexanediol failed to disperse the accumulation of OPTN and NDP52 on LC3-positive structures, suggesting that the condensate does not dissolve in the presence of isolation membranes. This may be because OPTN and NDP52 remain at the mitochondria-isolation membrane contact sites more stably due to the additional interaction with ATG8

proteins, not only with ubiquitin. These data are described on page 8, lines 171-174.

(b) Unlike 1,6-hexanediol, 2,5-hexanediol failed to disperse the accumulation of OPTN and NDP52 (Figure EV2B), as expected by the reviewer. The data were described on page 8, lines 168-170.

(c) NDP52 condensate dispersion was also evident after 1,6-hexanediol treatment in the absence of isolation membranes (Figure 2E). These data were described on page 7, line 167.

4. In Figure 3A, the authors assume that the adaptor proteins (OPTN) diffuse freely in solution to fill the cleft between mitochondria, which forms the basis of the mathematical model. However, in Figure EV1A, p62 also accumulates between clustered mitochondria, despite the FRAP results in Figure 2A showing slow recovery of p62 condensates. How is the mobility of these adaptors correlated with their actual functions? Further details should be included to further support the mathematical model.

We apologize for the lack of explanation and proper interpretation of the FRAP results with p62. Although p62 showed modest recovery with FRAP (Fig. 2B), it indeed accumulated between mitochondria (Figure EV1), as noted by the reviewer. In fact, we have observed that p62 also undergoes LLPS and exhibits fluidity. As shown in the Response Figure 3, fluorescence loss in photobleaching (FLIP) experiments conducted in clustered mitochondria revealed that GFP-p62 fluorescence diminished more rapidly when the detection area was in the same mitochondrial cluster (target area) than when it was in a separate mitochondrial cluster (control area). These data suggest that although the cytosol-condensate exchange is low, p62 displays high mobility within the condensates that cover the cluster of several mitochondria. Although we did not include the p62 FLIP data in the manuscript to maintain the focus on OPTN-dependent mitophagy, we have now discussed p62's liquidity on page 15, lines 362-368.

Response Figure 3. p62 shows fluidity on the surface of mitochondrial clusters.

(A) Schematic representation of the FLIP experiment. Photobleaching is applied to the “bleach” area (red square). The fluorescence of GFP-p62 or mRuby-Omp25 in the “control” and “target” areas were monitored to investigate molecular exchange between the “bleach” and “target” areas (blue arrows).

(B) Penta KO HeLa cells expressing GFP-p62 and mRuby-Omp25 were treated with CCCP for 45 min. Magnified time-lapse images of GFP-p62 in the “control”, “target”, and “bleach” areas are shown in the right panel. Scale bar, 4 μm .

(C) Quantification of fluorescence intensity GFP-p62 and mRuby-Omp25 in the “control”, “target” and “bleach” areas ($n \geq 4$). Error bars represent the SEM.

5. In Figure 5A, will the mobility of GFP-OPTN be affected by nanobody-mRuby-Ub, and further, mitophagy activity?

This is a very important point also raised by Reviewer #1, Comment #5. We have now added data of cells expressing GFP-OPTN (wild-type) along with nanobody-ubiquitin. Indeed, nanobody-Ub expression greatly enhanced its interaction also with wild-type GFP-OPTN (Fig. 5B). Additionally, it caused partial defects in both FRAP recovery and mitophagy, demonstrating an intermediate phenotype between cells expressing wild-type OPTN and those expressing nanobody-Ub and GFP-OPTN Δ UBAN (Figure 5C-E). We speculate that this intermediate phenotype is due to the presence of abundant endogenous ubiquitin, to which GFP-OPTN can also bind, and thereby the condensate formation was not completely abolished. We have now added the data and described them on page 11, lines 257-259 and 268-269.

Responses to the comments of Reviewer #3

In the manuscript by Yang et al., the authors probe the role of autophagy receptors in forming liquid condensates during Parkin-dependent mitophagy. Using a combination of cell biology and mathematical modelling, the authors find that OPTN and NDP52 associate with mitochondria in a dynamic fashion following mitophagy activation and redistribute their mitochondrial localization to sites of isolation membrane initiation. This is not seen with the other autophagy receptors, including p62. Further characterizing OPTN, the authors go on to show that strengthening its interaction with ubiquitin (via substituting the OPTN UBAN domain with an anti-GFP nanobody system) results in mitochondrial recruitment following stimulation but a loss in fluid-like dynamics and a failure to induce mitophagy. Finally, the authors demonstrate that the mitochondrially associated OPTN UBAN mutant does not recruit ATG9 to mitochondria to initiate mitophagy.

Overall this is a very interesting manuscript and potentially brings key knowledge about the role and mechanism that OPTN plays in initiating mitophagy. The authors have demonstrated the LLPS nature of OPTN on mitochondria, but I feel the key experimental data showing that this is critical for mitophagy need strengthening.

We would like to thank Reviewer #3 for the positive evaluation of our manuscript and for bringing up insightful questions. We have investigated the issues highlighted by the reviewer and addressed them accordingly.

Main Points:

1) In Figure 2E, as a control, does 1,6-Hexanediol affect p62 staining on mitochondria in a similar manner to OPTN? Given the results in panel A, is it more stable than OPTN?

As anticipated by the reviewer, p62 condensate showed greater resistance to 1,6-hexanediol treatment (Figure EV2A). We added the description on page 8, line 168.

2) The data in Figure 5 are key in defining OPTN LLPS as a key mechanism for

mitophagy. However, is it possible that the orientation/conformation of OPTN (upon mitochondrial recruitment with the nanobody) that is restricted and not LLPS per se? Does replacement of the UBAN domain with GFP at the C-terminus also block mitophagy?

We appreciate the reviewer's insightful question. According to the reviewer's suggestion, we replaced the UBAN domain of OPTN with GFP at the C-terminus (OPTN Δ UBN-GFP) and coexpressed it together with nanobody-mRuby-Ub. Although CCCP treatment resulted in clear recruitment of OPTN Δ UBN-GFP on mitochondria, both FRAP recovery and mitophagy were impaired (Fig. EV4A, B, E and F), as observed in cells expressing GFP-OPTN Δ UBN (N-terminal GFP). These new data support our previous assertion that the loss of mitophagy activity in these cells was indeed due to impaired LLPS rather than the altered orientation. We added the description on page 12, lines 285-290. Additionally, we discussed the possible contribution of the restricted orientation of these molecules due to stoichiometric interaction in comparison to flexible orientation in LLPS on page 12, lines 290-292.

3) Related to the above, loss of the C-terminal domain of OPTN, could be considered an extreme modification to the protein. Is this construct phosphorylated by TBK1 upon mitophagy initiation? Are similar observations also seen with just a point mutant of OPTN that impairs ubiquitin binding (such as the ALS-related E478G mutation)?

We appreciate this valuable suggestion. Indeed, ubiquitin binding-deficient OPTN (E478G) replicated the phenotype of OPTN Δ UBAN. OPTN (E478G) failed to localize on damaged mitochondria in the presence of wild-type ubiquitin, and nanobody-ubiquitin expression restored the recruitment of OPTN (E478G) (Figure EV4A). FRAP recovery was comparable between OPTN (E478G) and OPTN Δ UBAN when coexpressed with nanobody-ubiquitin (Figure EV4B). Moreover, OPTN (E478G) failed to induce mitophagy in the presence of nanobody-ubiquitin (Figure EV4C and D). These data suggest that the observed phenotype is indeed due to the lack of binding to ubiquitin (and forced interaction between GFP and nanobody), rather than unexpected effects such as conformational changes. We included these data and described them on page 12, lines 282-285.

Regarding the phosphorylation of OPTN by TBK1, we found that TBK1 recruitment to damaged mitochondria was impaired in cells expressing OPTN Δ UBAN and GFP-Ub (Figure EV5B). This is likely because the OPTN-TBK1 feed-forward mechanism, which was recently proposed by the Lazarou and Matsuda groups (Nguyen et al., Mol Cell 2023, Yamano et al., EMBO J 2024), is also defective. Therefore, we think that it would be difficult to evaluate OPTN phosphorylation properly in the absence of proper recruitment of TBK1 in this experiment. These new data are described on page 13, lines 321-331. We hope that the experiment using the OPTN (E478G) mutant sufficiently addresses this comment.

4) The data suggests that affinity of receptor for ubiquitin binding is important for mitochondrial redistribution to sites of isolation membrane initiation. Given that p62 is recruited to ubiquitinated mitochondria but does not redistribute, is anything known about the ubiquitin-binding affinities of p62's UBA domain compared to OPTN's UBAN domain: does substitution of the UBAN domain on OPTN with p62's UBA domain result in a block in OPTN redistribution and mitophagy?

We appreciate the reviewer's intriguing question. We expressed an OPTN mutant where the UBAN domain was replaced with p62's UBD domain. However, this mutant failed to localize on damaged mitochondria (and thus to induce mitophagy) in penta KO cells, and therefore, we were unable to test the possibility (Response Figure 4). We understand that it would be important to elucidate the molecular basis for the different mobility on the mitochondrial surface between OPTN/NDP52 and p62 in the future.

Response Figure 4. OPTN with the ubiquitin-binding domain of p62 fails to localize to mitochondria. (A) Localization of OPTN, whose UBAN domain was replaced with the ubiquitin-binding domain (UBD) of p62 in penta KO cells under mitophagy-inducing conditions (CCCP+, 60 min). Scale bar, 5 μ m. (B) Lysates from penta KO cells expressing the indicated OPTN and mutants were immunoblotted. Cells were treated with 1 μ M oligomycin and 2 μ M antimycin for 18 h.

Dear Noboru,

We have now received re-review reports from two referees, which I have included below. As you will see, both support publication. Before I can finally accept the manuscript though, there are some remaining editorial points which need to be addressed. In this regard would you please:

- remove the author credit section from the manuscript,
- either include data on p62 photobleaching on mitochondrial surfaces, or remove the sentence mentioning it (we no longer allow 'data not shown' references),
- upload individual, high-resolution figure files without legends (which should remain in ms file),
- remove instructions from the Reagents and Tools table,
- ZIP together all source data for EV and appendix figures,
- provide source data for the blots shown in Figure 5d and EV Fig 4: our software has detected a possible re-use between the HSP90-stained blots,
- provide legends for figures 1c-f are in a sequential manner (legends for figures 1e and 1f are currently provided before legend of figures 1c-d; 1d); this also goes for figure EV 1b-c (legend for figure 1c is provided before legend of figure 1b),
- define the p value in the legend of figure 6c,
- define white arrows in the legend of figure 1a-b, e; 4b, d, f; EV 1a, c, and
- remove movie legends from manuscript file and zip with each movie file.

We include a synopsis of the paper (see <http://emboj.embopress.org/>). Please provide me with a general summary image, two-sentence summary statement and 3-5 bullet points that capture the key findings of the paper.

I am looking forward to receiving your revised manuscript.

EMBO Press is an editorially independent publishing platform for the development of EMBO scientific publications.

Best wishes,

William

William Teale, PhD
Editor
The EMBO Journal
w.teale@embojournal.org

Please remember: Digital image enhancement is acceptable practice, as long as it accurately represents the original data and

conforms to community standards. If a figure has been subjected to significant electronic manipulation, this must be noted in the figure legend or in the 'Materials and Methods' section. The editors reserve the right to request original versions of figures and the original images that were used to assemble the figure.

We realize that it is difficult to revise to a specific deadline. In the interest of protecting the conceptual advance provided by the work, we recommend a revision within 3 months (3rd Dec 2024). Please discuss the revision progress ahead of this time with the editor if you require more time to complete the revisions. Use the link below to submit your revision:

Referee #1:

The authors have done a great job addressing the comments and strengthening the manuscript. The work makes an important advance toward our mechanistic understanding of selective autophagy mediated by autophagy adaptors. Congratulations on an excellent study.

Referee #3:

The authors have addressed all my concerns.

All editorial and formatting issues were resolved by the authors.

Dear Noboru,

I am pleased to inform you that your manuscript has been accepted for publication in the EMBO Journal.

Congratulations! I'm very happy to be able to publish this work.

Best wishes,

William

William Teale, PhD
Editor
The EMBO Journal
w.teale@embojournal.org
